# SARS-CoV-2 infection induces inflammatory bone loss in golden Syrian hamsters

Wei Qiao [1,2,3], Hui En Lau[1], Huizhi Xie[1,2], Vincent Kwok-Man Poon [4,5], Chris Chung-Sing Chan [4,5], Hin Chu [4,5,6], Shuofeng Yuan[4,5,6], Terrence Tsz-Tai Yuen[4], Kenn Ka-Heng Chik[4,5], Jessica Oi-Ling Tsang[4,5], Chris Chun-Yiu Chan[4], Jian-Piao Cai[4], Cuiting Luo[4], Kwok-Yung Yuen [4,5,6,7], Kenneth Man-Chee Cheung[1,2], Jasper Fuk-Woo Chan [4,5,6,7,8 ✉] & Kelvin Wai-Kwok Yeung [1,2 ✉]

Extrapulmonary complications of different organ systems have been increasingly recognized in patients with severe or chronic Coronavirus Disease 2019 (COVID-19). However, limited information on the skeletal complications of COVID-19 is known, even though inflammatory diseases of the respiratory tract have been known to perturb bone metabolism and cause pathological bone loss. In this study, we characterize the effects of severe acute respiratory syndrome coronavirus 2 (SARS-CoV-2) infection on bone metabolism in an established golden Syrian hamster model for COVID-19. SARS-CoV-2 causes significant multifocal loss of bone trabeculae in the long bones and lumbar vertebrae of all infected hamsters. Moreover, we show that the bone loss is associated with SARS-CoV-2-induced cytokine dysregulation, as the circulating pro-inflammatory cytokines not only upregulate osteoclastic differentiation in bone tissues, but also trigger an amplified pro-inflammatory cascade in the skeletal tissues to augment their pro-osteoclastogenesis effect. Our findings suggest that pathological bone loss may be a neglected complication which warrants more extensive investigations during the long-term follow-up of COVID-19 patients. The benefits of potential prophylactic and therapeutic interventions against pathological bone loss should be further evaluated.

[1] Department of Orthopaedics and Traumatology, School of Clinical Medicine, Li Ka Shing Faculty of Medicine, the University of Hong Kong, Hong Kong S.A.R., China. [2] Shenzhen Key Laboratory for Innovative Technology in Orthopaedic Trauma, the University of Hong Kong-Shenzhen Hospital, Shenzhen 518053, China. [3] Applied Oral Sciences & Community Dental Care, Faculty of Dentistry, the University of Hong Kong, Hong Kong S.A.R., China. [4] State Key Laboratory of Emerging Infectious Diseases, Carol Yu Centre for Infection, Department of Microbiology, School of Clinical Medicine, Li Ka Shing Faculty of Medicine, The University of Hong Kong, Pokfulam, Hong Kong Special Administrative Region, China. [5] Centre for Virology, Vaccinology and Therapeutics, Hong Kong Science and Technology Park, Hong Kong Special Administrative Region, China. [6] Department of Clinical Microbiology and Infection Control, The University of Hong Kong-Shenzhen Hospital, Shenzhen, Guangdong Province, China. [7] Academician Workstation of Hainan Province of Hainan Medical University, and Hainan Medical University-The University of Hong Kong Joint Laboratory of Tropical Infectious Diseases, Hong Kong Special Administrative Region, China. [8] Guangzhou Laboratory, Guangzhou, Guangdong Province, China. ✉email: jfwchan@hku.hk; wkkyeung@hku.hk

The severe acute respiratory syndrome coronavirus 2 (SARS-CoV-2) has caused nearly 346 million cases of Coronavirus Disease 2019 (COVID-19) and nearly 5.5 million deaths as of 23 January 2022 since the virus' discovery in December 2019[1]. Severe acute COVID-19 may be complicated by both pulmonary (pneumonia with acute respiratory distress syndrome and respiratory failure) and extrapulmonary manifestations, such as anosmia, ageusia, diarrhea, lymphopenia, and multi-organ dysfunction syndrome[2–4]. More recently, it has been increasingly recognized that some patients may develop long-term complications and persistent symptoms of COVID-19, such as fatigue, headache, dyspnea, anosmia, muscle weakness, low-grade fever, and cognitive dysfunction[5–7]. However, the full spectrum of clinical manifestations in the long-term post-acute sequelae of SARS-CoV-2 infection, or "long COVID", remains incompletely understood. In particular, SARS-CoV-2-associated pathological changes in the skeletal system remain largely unknown.

Recently, a multi-center study showed that COVID-19 patients requiring intensive care had significantly lower bone mineral density (BMD) than those who were managed in non-intensive care settings[8]. Another clinical study found that the number of severe clinical incidence was significantly higher in patients with lower BMD compared to those with higher BMD, therefore vertebral BMD is a strong independent predictor of mortality in COVID-19 patients[9]. In addition, about 24% of long COVID patients reported bone ache or burning, with the symptoms lasting for up to 7 months after the onset of COVID-19[10]. Despite these emerging evidence on the long-term complications of COVID-19, very limited serial investigations have been conducted on skeletal system involvement in the post-recovery phase. This is not unexpected because in COVID-19, patients either succumb or recover from the acute phase. In patients who recover from COVID-19, the focus on follow-up is usually limited to the respiratory, cardiac, and neurological functions which are well reported in the literature, rather than skeletal pathologies which typically do not manifest in the acute phase and therefore may be neglected. Moreover, since severe COVID-19 is most often found in elderly patients and those with comorbidities, including those on chronic corticosteroid and immunosuppressive drugs, the disease-associated bone changes in these patients who may already have osteoporosis before the infection may not be appreciated.

The skeletal system undergoes continuous bone formation and degradation throughout life and this tightly regulated remodeling process can be disturbed by many factors, particularly metabolic alterations and hormonal changes[11,12]. An imbalance between bone formation and resorption can also result from various chronic inflammatory diseases, leading to systemic osteoporosis and increased fracture risk[13,14]. For instance, chronic pulmonary inflammation arising from chronic obstructive pulmonary disease (COPD), cystic fibrosis, and asthma were reported to induce systemic bone loss[15,16]. Indeed, it has been shown that the extent of local or systemic osteopenia is associated with the degree of inflammatory response and the inflammation-induced bone loss can continue after effective therapeutic intervention on the inflammatory disease[13,17,18]. Severe COVID-19 patients developed much higher serum concentrations of pro-inflammatory cytokines and chemokines (e.g., IL-1β, IFN-α, IL-1RA, and IL-8) than those with milder disease, indicating that the cytokine dysregulation was closely correlated with disease severity[19–21]. The pro-inflammatory cytokines are known to not only perpetuate the inflammation to impair lung function but also perturb bone metabolism, leading to bone resorption[18,22]. A recent radiological study on COVID-19 survivors with persisting symptoms for up to three months after discharge revealed that the inflammation in bone marrow persisted after recovery[23]. Additionally, reactive hemophagocytosis mediated by cytokine dysregulation-induced activation of macrophages was common in deceased COVID-19 patients[24]. Therefore, we hypothesize that, in addition to other reported extrapulmonary manifestations, the cytokine dysregulation in COVID-19 patients may also contribute to pathological changes in the skeletal system[2].

In this work, we characterize the effects of SARS-CoV-2 infection on bone metabolism during the acute and post-recovery phases in our established golden Syrian hamster model which closely mimics human infection[25]. Moreover, we demonstrate that the inflammation-induced osteoclastic activation following the cytokine dysregulation plays an important role in pathological bone loss. The findings of this study highlight the need for optimizing clinical protocols for monitoring long-term complications of COVID-19 and finding novel treatment strategies for SARS-CoV-2-induced inflammatory osteopenia/osteoporosis.

## Results

**SARS-CoV-2 causes significant loss of bone trabeculae.** To investigate the in vivo effects of SARS-CoV-2 infection on bone metabolism, we utilized our established golden Syrian hamster model, which recapitulates the clinical, virological, immunological, and pathological features of COVID-19 in humans[25] (Fig. 1a, b). In this hamster model, the most prominent disease manifestations were seen at about 4 days post-infection (dpi) and the hamsters generally recovered at about 7 to 10 dpi. As shown in our three-dimensional micro computerized tomography (μCT) scans, SARS-CoV-2-infected but not PBS-treated hamsters exhibited progressive loss of bone trabeculae at the distal metaphysis of femurs from the acute phase (4 dpi) to the post-recovery phase (30 dpi) and the chronic phase (60 dpi) of infection (Fig. 1c). A significant decrease in the trabecular bone volume fraction, bone mineral density, trabecular thickness, and the trabecular number could be evidenced at as early as 4 dpi (Fig. 1d and Supplementary Fig. 1a). There was also a significant decrease in the polar moment of inertia, though the cortical bone area and cortical thickness were barely affected (Supplementary Fig. 1b). On day 60, when the hamsters had recovered from the SARS-CoV-2 infection, the bone density was not restored. Instead, a gradual decrease of bone volume fraction, as well as progressive increases of trabecular pattern factor and the specific bone surface was detected (Fig. 1d). To verify our μCT scan findings, we examined the histological changes in the femurs of the hamsters (Fig. 1e and Supplementary Fig. 2a) and confirmed the reduction in bone trabeculae structures at the distal metaphysis of the femur was evident in the SARS-CoV-2-infected but not the mock-infected hamsters.

Next, we investigated whether similar changes were present in other sites. Both the μCT scan data and histological analysis showed that there was also a significant decrease in bone trabeculae at the proximal metaphysis of the tibia (Fig. 2a–c). The trabecular bone volume of SARS-CoV-2-infected hamsters at 60 dpi was less than 50% of that of the mock-infected hamsters (Fig. 2b). Corroboratively, there were significant decreases in bone mineral density, trabecular thickness, trabecular number, and polar moment of inertia after SARS-CoV-2 infection. Similar to our findings in the femur, the trabecular pattern factor, trabecular separation, and specific bone surface progressively decreased throughout the period of observation (Fig. 2b and Supplementary Fig. 1c). The same pattern of bone loss with a significant reduction in trabecular bone volume fraction, bone mineral density, trabecular thickness, trabecular number, and polar moment of inertia were observed also in the lumbar vertebrae at 30 dpi (Fig. 2d, e and Supplementary Fig. 1d). Meanwhile, a higher trabecular pattern factor and specific bone

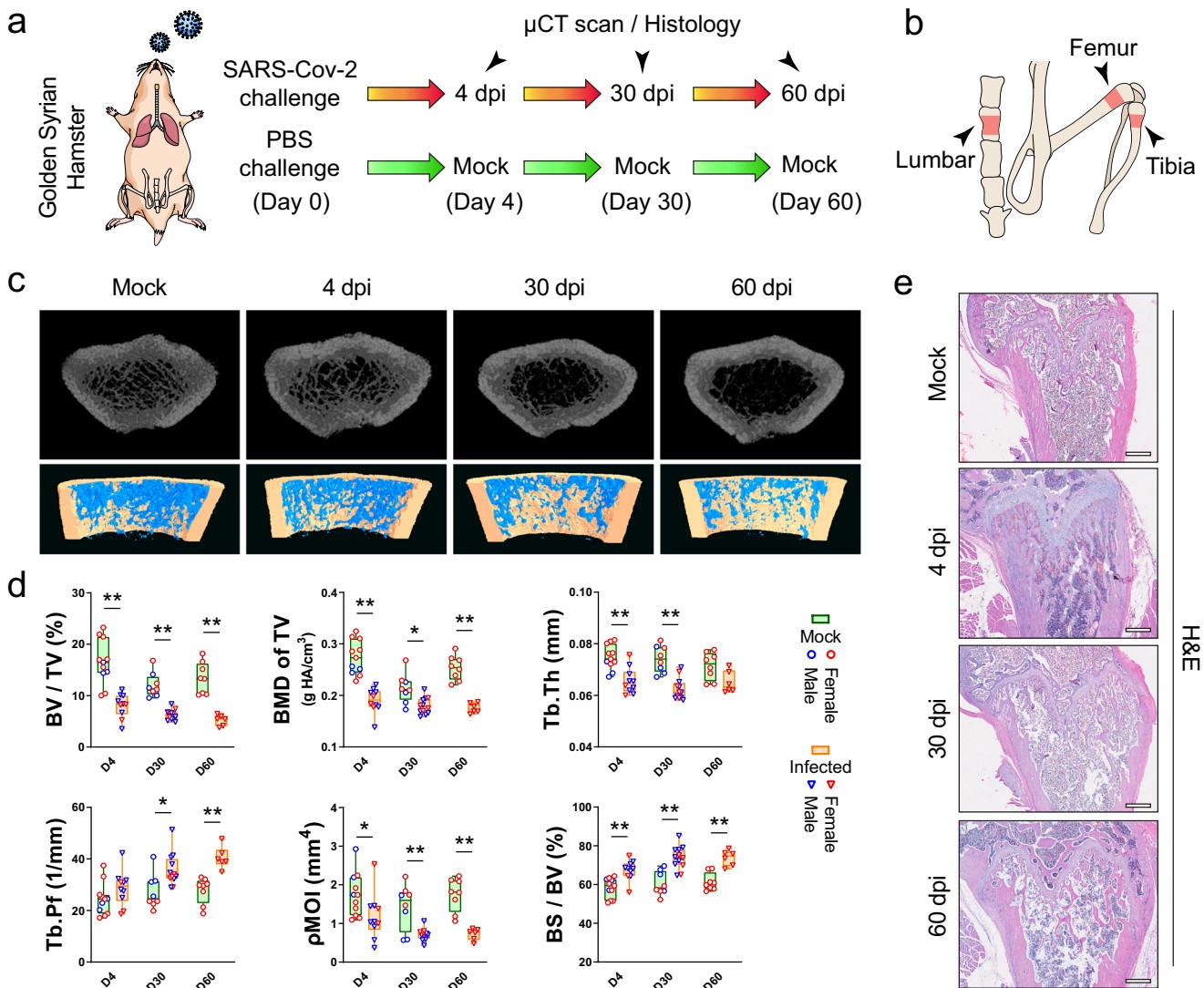

**Fig. 1 SARS-CoV-2 induces bone loss in the golden Syrian hamster model. a** Golden Syrian hamsters were either treated with SARS-CoV-2 or PBS (Mock), followed by a μCT scan and histology analysis at 4, 30, and 60 days post-infection (dpi). **b** The regions of interest for μCT evaluation of bone density included the distal metaphysis of the femur, proximal metaphysis of tibia, and lumbar vertebrae. **c** Representative μCT images showing the reduction in trabecular bone volume in the femurs of the SARS-CoV-2-infected hamsters. **d** Corresponding measurements of trabecular bone volume fraction (BV/TV), bone mineral density (BMD of TV), trabecular thickness (Tb.Th), trabecular pattern factor (Tb.Pf), polar moment of inertia (pMOI), and specific bone surface (BS/BV). Mock: $n = 11$ (D4), 8 (D30), 8 (D60); Infected: $n = 10$ (D4), 12 (D30), 6 (D60). **e** Representative H&E staining images showing the cancellous bone structures in the femurs of the SARS-CoV-2-infected hamsters ($n = 4$, scale bars = 500 μm). Data were presented as box plots with whiskers from minima to maxima, the central line at the 50th percentile, and the ends of the box at the 25th and 75th percentiles. *$P < 0.05$, **$P < 0.01$ by two-way ANOVA with Bonferroni's multiple comparisons test.

surface were detected after the infection. Overall, these findings showed that the SARS-CoV-2 infection causes a significant bone loss at different sites of a skeleton in the hamster model.

**SARS-CoV-2 activates osteoclastogenesis in hamsters**. To provide mechanistic insights into the dysregulated bone metabolism in SARS-CoV-2-infected hamsters, we asked whether the bone loss is primarily caused by an alteration in bone resorption or in bone formation. Compared with mock-infected hamsters, a significantly higher number of tartrate-resistant acid phosphatase-positive (TRAP+) osteoclasts were found in the bone trabeculae at the distal metaphysis of the femur (Fig. 3a), the proximal metaphysis of the tibia (Fig. 3b), and the lumbar vertebrae (Fig. 3c) of SARS-CoV-2-infected hamsters. The number of TRAP+ osteoclasts at the distal metaphysis of the femur of the

SARS-CoV-2-infected hamsters at 4 dpi and 30 dpi was almost double that of the mock-infected hamsters (Fig. 3d). Moreover, immunofluorescence staining demonstrated significantly more TRAP+ osteoclasts expressing nuclear factor of activated T-cells, cytoplasmic 1 (NFATc1) at the bone surface of SARS-CoV-2-infected hamsters (Fig. 3e). The increased intensities of TRAP and NFATc1 at the distal metaphysis of the femur at 4 dpi indicated that the osteoclastic activity was upregulated by SARS-CoV-2 infection (Fig. 3f). We also compared the expression of NFATc1, TRAP, cathepsin K, and receptor activator of NF-κB (RANK) in the bone tissues using Western blotting, which showed the upregulation of these osteoclastic markers after SARS-CoV-2 infection (Supplementary Fig. 2b, c). In contrast, alkaline phosphatase (ALP) staining (Supplementary Fig. 3a) and immunofluorescence staining for osteocalcin (Supplementary Fig. 3b) showed that there was no significant difference in the

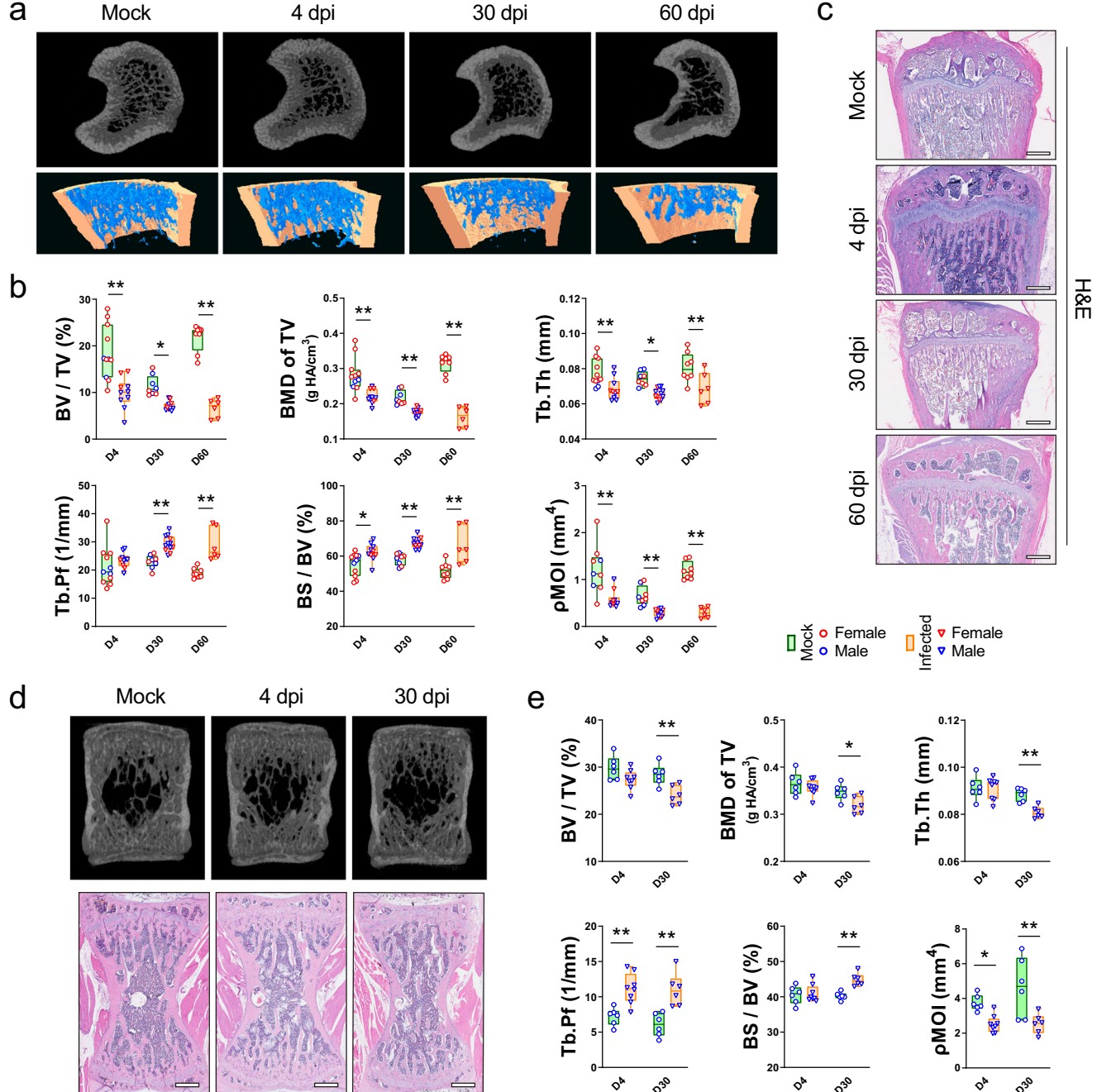

**Fig. 2 Bone loss in the tibias and lumbar vertebrae after SARS-CoV-2 infection in the golden Syrian hamster model. a** Representative μCT image showing the reduction in trabecular bone volume in tibias of the SARS-CoV-2-infected hamsters. **b** Corresponding measurements of BV/TV, BMD of TV, Tb.Th, Tb.Pf, BS/BV, and pMOI. Mock: $n = 11$ (D4), 8 (D30), 8 (D60); Infected: $n = 10$ (D4), 12 (D30), 6 (D60). **c** Representative H&E staining images showing the cancellous bone structures in tibias of the SARS-CoV-2-infected hamsters ($n = 4$, scale bars = 500 μm). **d** Representative μCT and corresponding H&E staining images showing the bone loss in the lumbar vertebrae of the SARS-CoV-2-infected hamsters. **e** Corresponding measurements of BV/TV, BMD of TV, Tb.Th, Tb.Pf, BS/BV, and pMOI. Mock: $n = 6$ (D4), 6 (D30); Infected: $n = 8$ (D4), 6 (D30). Data were presented as box plots with whiskers from minima to maxima, the central line at the 50th percentile, and the ends of the box at the 25th and 75th percentiles. *$P < 0.05$, **$P < 0.01$ by two-way ANOVA with Bonferroni's multiple comparisons test.

osteoblastic activities between the SARS-CoV-2-infected and mock-infected hamsters at 4 dpi.

We next determined the expression of various osteoclastogenesis-related genes in the bone tissue of SARS-CoV-2-infected hamsters. Compared with mock-infected hamsters, the expression of receptor activator of nuclear factor-kappa B ligand (*RANKL*), which is essential for the osteoclastic differentiation, was tripled in SARS-CoV-2-infected hamsters (Fig. 4a). Moreover, several osteoclastic

marker genes contribute to the formation and activity of osteoclasts, including receptor activator of NF-κB (*RANK*), cathepsin K (*CTSK*), matrix metallopeptidase 9 (*MMP-9*), and colony-stimulating factor 1 receptor (*CSF1R*), were significantly upregulated in the bone tissues after SARS-CoV-2 infection (Fig. 4a). The expression of osteoprotegerin (*OPG*), which acts as a decoy receptor for RANKL to inhibit RANK-RANKL mediated osteoclastogenesis and bone resorption, was correspondingly significantly

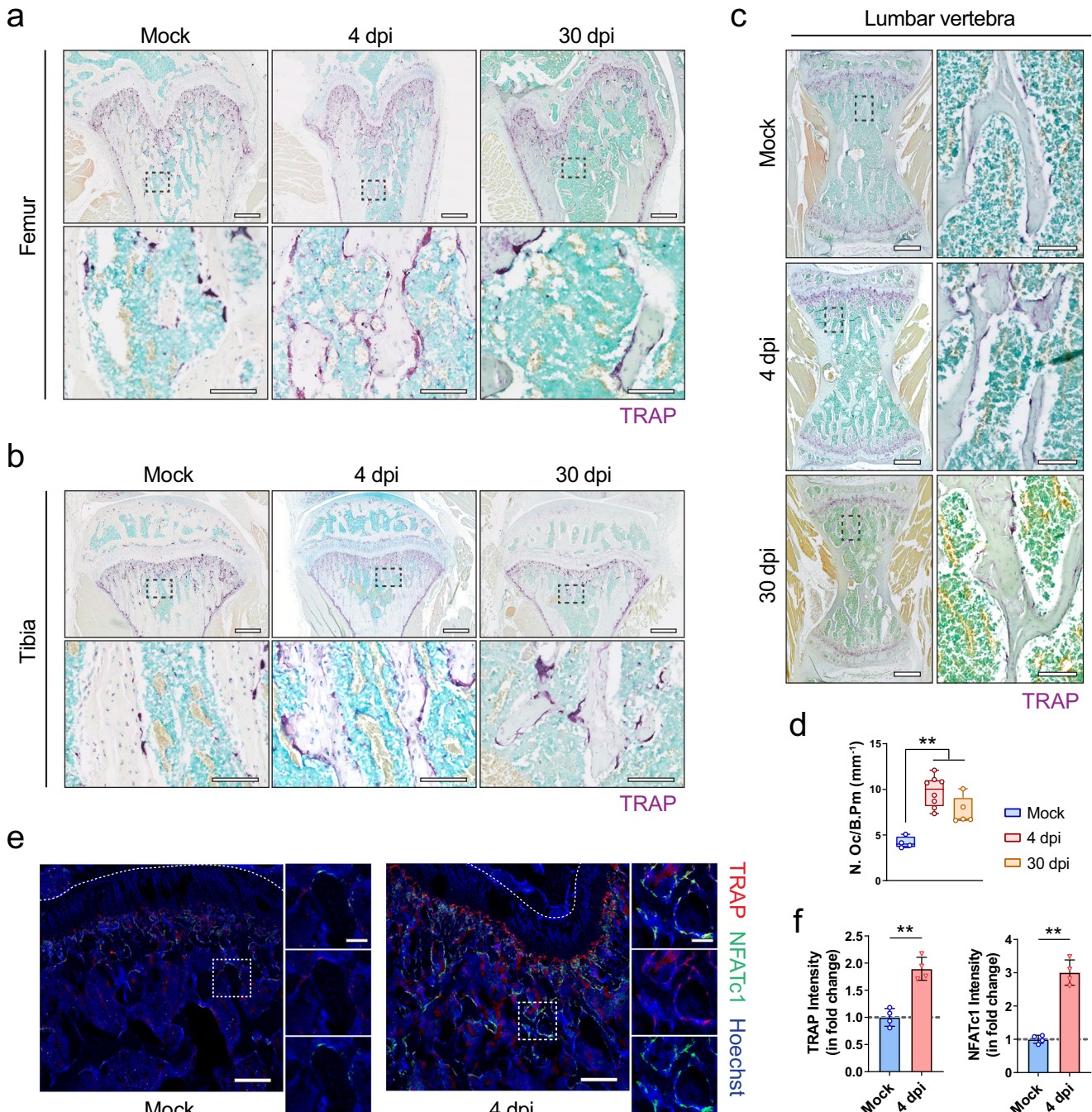

**Fig. 3 Osteoclastogenesis in the femurs and tibias of the hamsters after SARS-CoV-2 infection. a–c** Representative TRAP staining showing the increase in the number of TRAP+ osteoclasts at **a** the distal metaphysis of the femur ($n = 4$), **b** the proximal metaphysis of the tibia ($n = 4$), and **c** the lumbar vertebrae ($n = 4$) after the treatment with PBS (Mock) or SARS-CoV-2 infection (4 dpi). Lower images (scale bars = 100 μm) are high-resolution versions of the boxed regions in the upper images (scale bars = 500 μm). **d** Corresponding quantification of TRAP+ osteoclasts at the trabecular bone surface after treatment with PBS (Mock, $n = 4$) or SARS-CoV-2 infection (4 dpi, $n = 8$, 30 dpi, $n = 5$). **e** Representative immunofluorescence staining images and **f** the corresponding fluorescence intensity quantification showing the increase in the number of TRAP+ NFATc1+ osteoclasts at the distal metaphysis of the femur at 4 dpi ($n = 4$). Tile scans (scale bars = 200 μm) of the distal femoral metaphysis are shown along with high-magnification of the boxed regions (scale bars = 50 μm). Data were mean ± SD. ns: $P > 0.05$, *$P < 0.05$, **$P < 0.01$ by one-way ANOVA with Tukey's post hoc test (**d**) or two-sided Student's $t$-test (**f**).

downregulated after the infection (Fig. 4a). In addition to the increase in the number of TRAP+ osteoclasts, we also found that there were significantly more osteoclast progenitors, including CD68+ macrophages and RANK+ preosteoclasts, after SARS-CoV-2 infection (Fig. 4b, c). There were more multinuclear cells co-expressing TRAP and CD68 located at the bone trabeculae of

SARS-CoV-2-infected hamsters (Fig. 4b). Using multiplex immunohistochemical (IHC) staining, we further identified the upregulated lineage of TRAP+RANK+CD68+ osteoclasts that expressed a higher level of interleukin-1 beta (IL-1β) in SARS-CoV-2-infected hamsters than mock-infected hamsters (Fig. 4d). Thus, to further elucidate the underlying mechanism, we then examined the

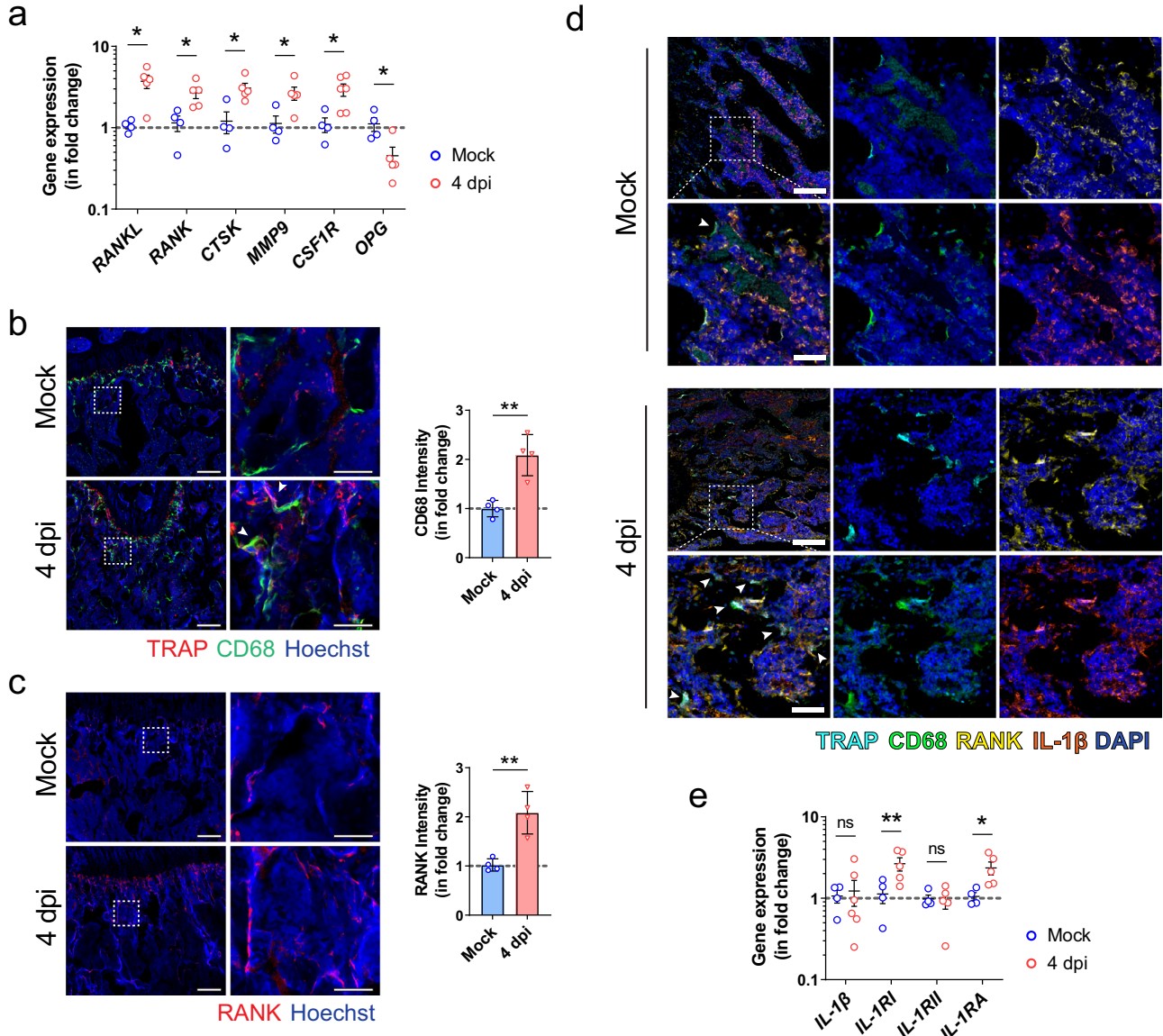

**Fig. 4 The inflammatory activation and osteoclastic differentiation of monocyte-macrophage lineage after SARS-CoV-2 infection. a** The expression of osteoclastogenesis-related genes in the bone tissue of the hamsters at day 4 after treatment with PBS (Mock, $n = 4$) or SARS-CoV-2 infection (4 dpi, $n = 5$). **b** Representative immunofluorescence staining images showing the increase in the number of CD68$^+$ and TRAP$^+$ osteoclasts (indicated by arrowhead) after SARS-CoV-2 infection ($n = 4$). Tile scans (scale bars = 200 μm) and high-magnification of the boxed regions (scale bars = 50 μm), as well as corresponding quantification for the fluorescence intensity of CD68, are shown. **c** Representative immunofluorescence staining images and the corresponding quantification showing the upregulation of RANK-expressing cells at the trabecular bone surface after SARS-CoV-2 infection. Tile scans (scale bars = 200 μm) and high-magnification of the boxed regions (scale bars = 50 μm) are shown. **d** Representative multicolor immunohistochemical staining for TRAP, CD68, RANK, and IL-1β was performed at the distal metaphysis of the femur on day 4 after treatment with PBS (Mock) or SARS-CoV-2 infection (4 dpi). DAPI was used for nuclear counterstaining. Tile scans (scale bars = 200 μm) and high-magnification of the boxed regions (scale bars = 50 μm) are shown. Arrowheads indicate osteoclasts on the bone surface ($n = 3$). **e** The expression of IL-1β signaling-related genes in bone tissue at day 4 after treatment with PBS (Mock, $n = 4$) or SARS-CoV-2 infection (4 dpi, $n = 5$). Data were mean ± SD. ns: $P > 0.05$, *$P < 0.05$, **$P < 0.01$ by two-sided Student's t-test.

expression of genes related to IL-1β signaling. Compared with the mock-infected hamsters, SARS-CoV-2-infected hamsters exhibited significant upregulation of interleukin-1 receptor type I (*IL-1RI*), but not interleukin-1 receptor type II (*IL-1RII*). Interestingly, the expression of IL-1β in the bone tissue was not significantly different between the two groups (Fig. 4e). Moreover, SARS-CoV-2 infection also induced a twofold increase in the expression of interleukin-1 receptor antagonist (*IL-1RA*) in bone tissue. Taken together, these findings indicate that SARS-CoV-2 infection leads to the activation of osteoclastic cascade resulting in the destruction of trabeculae structure in both long bones and axial skeleton.

**SARS-CoV-2 disturbs inflammatory microenvironment in skeleton system without direct infection.** Having demonstrated the involvement of the local immune response in the pathological osteoclastogenesis in SARS-CoV-2 infection, we next asked whether the inflammatory bone loss was also caused by direct SARS-CoV-2 infection of the bone tissue. In SARS-CoV-2-infected hamsters, viral nucleocapsid protein (NP) in co-localized angiotensin-converting enzyme 2 (ACE2)-expressing cells were evident throughout the respiratory tract, from the nasal turbinate to the trachea and pulmonary alveoli at 4 dpi (Supplementary Fig. 4). CD68$^+$ macrophages engulfing SARS-CoV-2-infected

cells, which co-expressed viral NP and ACE2, were also observed, indicating the presence of an active immune response in these areas. In stark contrast, despite the presence of ACE2 in some of the immune cells residing in the bone tissue, viral NP was absent in the periosteum, bone trabeculae, and synovium of the femoral bone tissue in the SARS-CoV-2-infected hamsters (Fig. 5a). Moreover, viral RNA was not detected in the bone tissue (Fig. 5b). These findings indicated that the bone tissues were not directly infected by SARS-CoV-2. We then investigated whether osteoclastogenesis was associated with the virus-induced inflammatory response. Our ELISA results showed that at 4 dpi the mean serum IL-1β, tumor necrosis factor-alpha (TNF-α), and interleukin 6 (IL-6) protein levels were all higher in the SARS-CoV-2-infected hamsters than that of the mock-infected hamsters, with the levels of IL-1β and TNF-α reaching statistical significance ($P < 0.01$) (Fig. 5c).

Interestingly, while direct SARS-CoV-2 infection of the bone tissue was absent, the expression of interferon-gamma (IFN-γ) and its downstream signals, including interferon regulatory factor 1 (IRF1) and interferon regulatory factor 2 (IRF2), were significantly upregulated in the bone tissue of SARS-CoV-2-infected hamsters at 4 dpi (Fig. 5d). The expression of interferon-induced protein with tetratricopeptide repeats 3 (IFIT3) was only marginally increased. Compared with the mock control, the inflammation-related genes upregulated in bone tissue in response to the respiratory infection of SARS-CoV-2 include C-C motif chemokine 22 (CCL22), interleukin-2 receptor antagonist (IL-2RA), TNF-α, colony-stimulating factor 1 (CSF1), and colony-stimulating factor 2 (CSF2). Nevertheless, the expressions of C-C motif chemokine 17 (CCL17), C-X-C motif chemokine ligand 10 (CXCL10), and interleukin 12 p40 (IL12p40) remained similar (Fig. 5d, e). Additionally, it is noteworthy that the infection in the respiratory system intriguingly downregulated the expression of interferon-beta (IFN-β), interleukin 21 (IL-21), and IL-6 in the bone tissue, which are vital for the clearance of virus infection.

**SARS-CoV-2-induced inflammatory cytokines upregulate osteoclastogenesis.** To address whether the bone loss subsequent to SARS-CoV-2 infection was caused by the pro-inflammatory cytokines originating from the respiratory system, we first compared the expression of various inflammatory cytokines in the bone tissue of SARS-CoV-2-infected and mock-infected hamsters. Using immunofluorescence staining, we showed that SARS-CoV-2 infection contributed to a significant increase in the levels of IL-1β, IL-1RA, and TNF-α, as well as a marginal increase in the level of IFN-γ in the bone tissue (Fig. 6a–d). These changes were confirmed by Western blotting which demonstrated a more than a sixfold increase of IL-1β, a sevenfold increase of TNF-α, and a threefold increase of IL-1RA in the bone tissue at 4 dpi in the SARS-CoV-2-infected hamsters compared to the mock-infected hamsters (Supplementary Fig. 5a, b). Additionally, immunofluorescence staining showed the co-expression of these inflammatory cytokines with CD68, the marker for bone resident macrophage (Fig. 6a–d). More importantly, we showed that the expression of NF-κB, the key transcription factor in inflammatory responses, was significantly higher at the bone surface of SARS-CoV-2-infected hamsters than that of the mock-infected hamsters (Fig. 6e). Semi-quantification by both immunofluorescence staining and Western blotting showed that the level of NF-κB in the bone tissue of SARS-CoV-2-infected hamsters was doubled that of the mock-infected hamsters (Fig. 6a and Supplementary Fig. 5a, b).

To further confirm the effects of these inflammatory cytokines on osteoclastic activities in bone tissue, we conducted a series of experiments using mouse bone marrow macrophages (BMMs) isolated from young (three-month-old) or matured adult (six-month-old) mice. First, we tested the response of murine BMMs to the stimulation of IL-1β as our previous experiment showed the activation of the IL-1β signaling cascade in SARS-CoV-2-infected hamsters (Fig. 4e). The presence of recombinant murine IL-1β contributed to a twofold increase in the number of TRAP⁺ multinuclear cells derived from BMM isolated from young mice (Fig. 7a, b). When added to BMMs isolated from matured adult mice, IL-1β led to a doubled size of fused osteoclasts (Fig. 7a, b). In contrast, the addition of IL-1β neutralizing antibody (Neu-Ab) not only reduced the number of BMM-derived osteoclasts but also resulted in a smaller size of osteoclasts differentiated from matured adult mouse BMMs. Meanwhile, the expression of IL-1R1 was increased by six times in the young BMMs and four times in the matured adult BMMs (Fig. 7c). However, the effect of Neu-Ab on the downregulation of IL-1R1 was only significant in BMMs from young mice, but not the ones from matured adult mice. Nevertheless, the gene expression of IL-1RA, which encodes interleukin-1 receptor antagonist, was significantly upregulated in both young and matured adult BMMs in response to the stimulation of IL-1β and downregulated after the addition of Neu-Ab.

Besides the direct effect of IL-1β on BMMs, we further tested whether IL-1β could promote osteoclastogenesis through the regulation of the osteoblast lineage. The indirect coculture of BMMs with mesenchymal stem cells (MSCs) was achieved using a transwell assay. IL-1β led to a significant increase in the size of BMM-derived osteoclasts after their coculture with IL-1β-treated MSCs, even though the number of TRAP⁺ multinuclear cells remained unchanged (Fig. 7d, e). Neu-Ab-treated MSCs, instead, decreased the average size of osteoclasts. This might be explained by the changes in the pro-osteoclastogenesis cytokines produced by MSCs because IL-1β contributed to a twofold increase in the expression of RANKL without changing the level of OPG in MSCs (Fig. 7f). Neither IL-1β nor its neutralizing antibody significantly regulated the expression of CSF1 in MSCs.

The effects of IL-1β and its neutralizing antibody on osteoclastogenesis were further confirmed using Western blotting. In both osteoclasts derived from BMM of young and matured adult mice, the expressions of NFATc1, NF-κB p65, and CTSK were significantly upregulated by recombinant murine IL-1β (Fig. 8a, b). Additionally, Neu-Ab suppressed the pro-osteoclastogenesis effects of IL-1β, as it downregulated the expression of NFATc1, NF-κB p65, and CTSK in BMM-derived osteoclasts from young and matured adult mice. Moreover, the Neu-Ab inhibited the IL-1β induced phosphorylation of c-Jun N-terminal kinase (JNK) (Supplementary Fig. 5c). Similar to the direct effect of IL-1β on osteoclastogenesis, IL-1β-treated MSCs also led to a significant upregulation in the expression of NFATc1 and the phosphorylation of JNK, which were both significantly downregulated by Neu-Ab-treated MSCs (Supplementary Fig. 6a). Although IL-1β-treated MSCs only contributed to a marginal increase in the expression of NF-κB p65 and CTSK in BMMs, Neu-Ab-treated MSCs significantly inhibited both of them (Supplementary Fig. 6a).

**The pro-inflammatory cascade promotes pathological bone resorption.** Since our results have verified the absence of SARS-CoV-2 infection in bone tissue, we hypothesized that the prominent inflammatory response in bone tissue was related to the circulating inflammatory cytokines that originate from the respiratory system after viral infection. Therefore, in addition to the pro-osteoclastogenesis effect of IL-1β, we also explored the immunomodulatory effects of IL-1β and its neutralizing

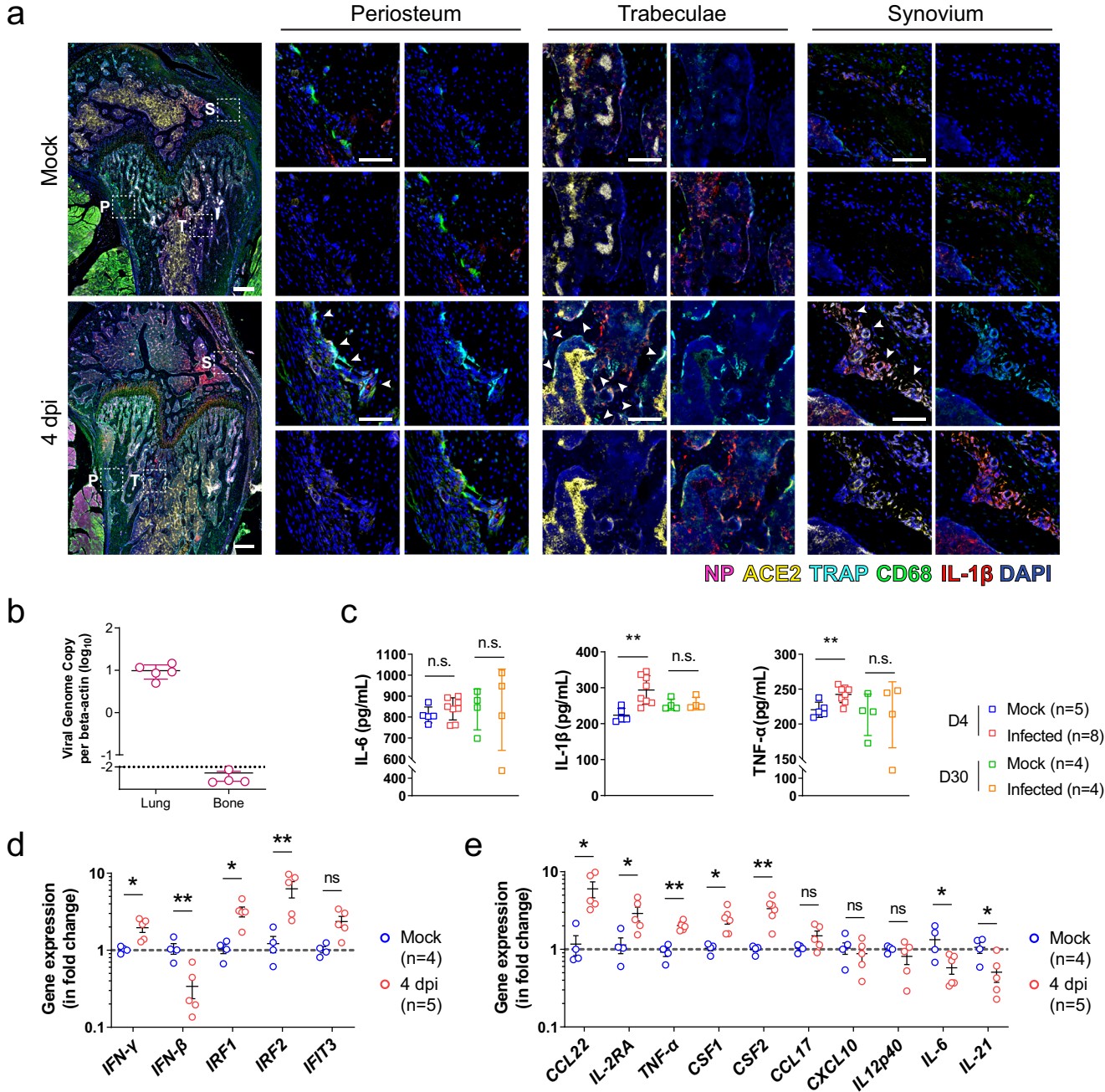

**Fig. 5 The absence of SARS-CoV-2 infection in bone tissues. a** Representative multicolor immunohistochemical staining for SARS-CoV-2 nucleocapsid protein (NP), angiotensin-converting enzyme 2 (ACE2), TRAP, CD68, and IL-1β was performed at the distal metaphysis of the femur after challenge with either PBS (Mock) or SARS-CoV-2 (4 dpi). Tile scans (scale bars = 200 μm) and high-magnification of the boxed regions (P periosteum, T trabeculae, S synovium; scale bars = 50 μm) are shown. Arrowheads indicate osteoclasts on the bone surface (n = 3). **b** Viral genome copies of the lung tissue (n = 5) and bone tissue (n = 4) harvested from SARS-CoV-2-infected hamsters. The viral genome copy was below the detection limit (dashed line) for all four bone samples. **c** The inflammatory cytokines, including IL-6, IL-1β, and TNF-α, in the serum of the hamsters treated with either PBS or SARS-CoV-2. **d** The expression of interferon signaling-related genes and **e** viral infection-associated inflammatory genes in bone tissue at day 4 after treatment with either PBS or SARS-CoV-2. Data were mean ± SD. ns: $P > 0.05$, $*P < 0.05$, $**P < 0.01$ by two-way ANOVA with Bonferroni's post hoc test (**c**) or two-sided Student's $t$-test (**d**, **e**).

antibody on BMMs isolated from young and matured adult mice. The recombinant murine IL-1β contributed to an approximately sixfold increase in the expression of cyclooxygenase-2 (COX2) and prostaglandin E synthase (PTGES) (Fig. 8b), which are both involved in the synthesis of prostaglandin E2 (PGE2). The presence of Neu-Ab inhibited the increase in the expression of COX2 and PTGES induced by IL-1β. The pro-inflammatory effect of IL-1β was also manifested by

the significant changes in the expression of various inflammation-related genes. For instance, in BMMs isolated from young mice, IL-1β led to a more than 80-fold increase in IFN-γ expression and an around twofold increase in TNF-α expression, as well as a fivefold increase in IL-6 expression. In contrast, the expressions of IL-10 and IL-23 were significantly downregulated by IL-1β, with the expression of IL-22 remained unchanged (Fig. 8c and Supplementary Fig. 6b). In BMMs

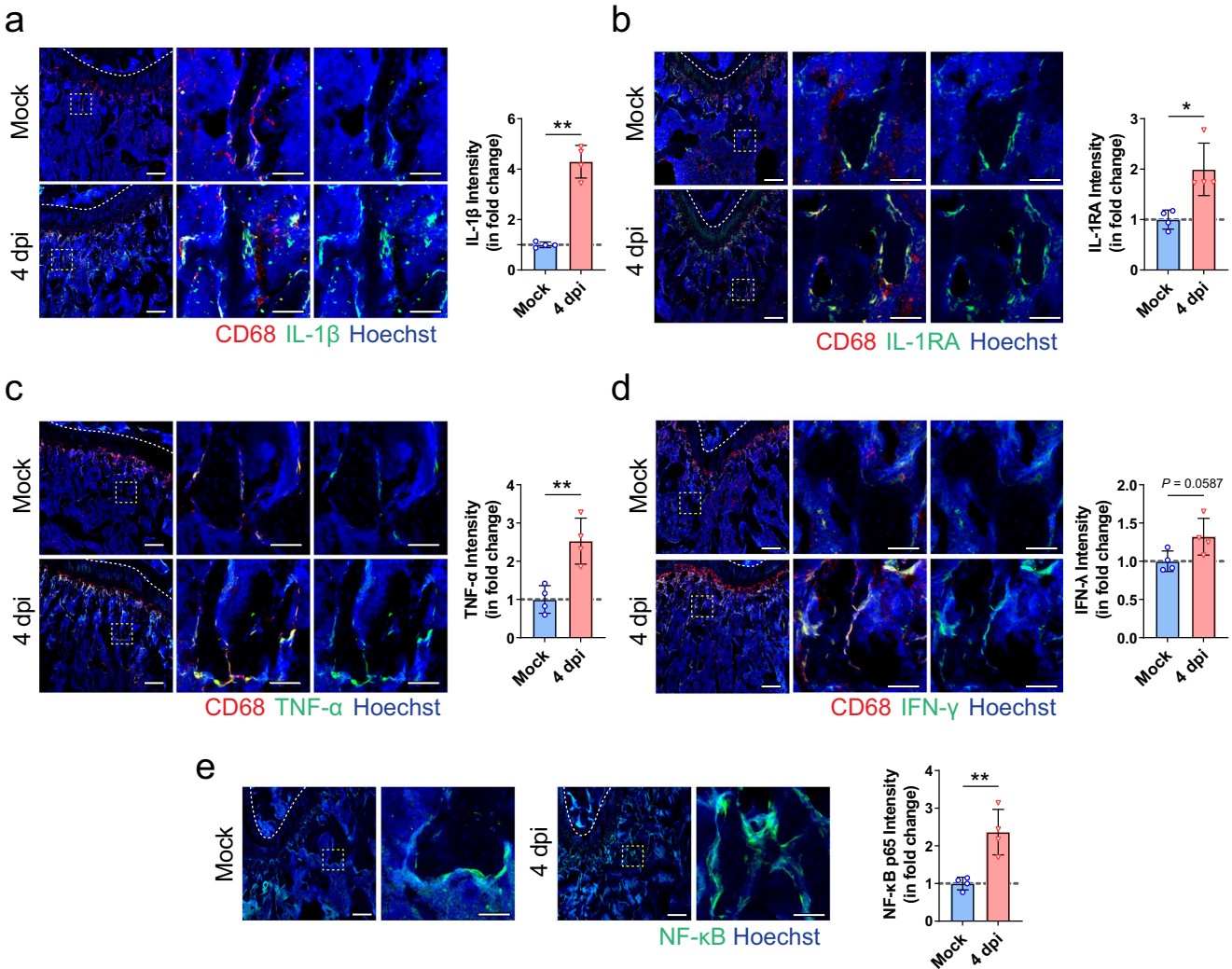

**Fig. 6 Macrophage-mediated inflammatory response in bone tissues after SARS-CoV-2 infection. a–d** Representative immunofluorescence staining images and the corresponding quantification showing the increase in the expressions of **a** IL-1β, **b** IL-1RA, **c** TNF-α, and **d** IFN-γ, at the distal metaphysis of the femur after treatment with PBS (Mock, n = 4) or SARS-CoV-2 infection (4 dpi, n = 4). Tile scans (scale bars = 200 μm) and high-magnification of the boxed regions (scale bars = 50 μm) are shown. **e** Representative immunofluorescence staining images and the corresponding quantification showing the increase in NF-κB p65 expression at the distal metaphysis of the femur after treatment with PBS (Mock, n = 4) or SARS-CoV-2 infection (4 dpi, n = 4). Data were mean ± SD. *P < 0.05, **P < 0.01 by two-sided Student's t-test.

isolated from matured adult mice, IL-1β also contributed to a 60-fold increase in *IFN-γ* expression, an approximately 1.75-fold increase in *TNF-α* expression, and an around fourfold increase in *IL-6* expression, and a less than 1.5-fold increase in *IL-22* expression. However, the expression of *IL-10* and *IL-23* were not significantly altered by the addition of IL-1β (Fig. 8c and Supplementary Fig. 6b). In BMMs from both young and matured adult mice, the addition of Neu-Ab abolished the upregulation in the expression of *IFN-γ*, *TNF-α*, and *IL-6* caused by IL-1β. In BMM from young mice, Neu-Ab resulted in significant upregulation in the expression of *IL-10*, *IL-22*, and *IL-23* (Fig. 8c and Supplementary Fig. 6b).

We then asked whether the amplified and broadened pro-inflammatory cascade in bone following the SARS-CoV-2 infection-induced cytokine dysregulation synergistically promoted osteoclastogenesis. First, we showed that recombinant murine TNF-α and IFN-γ exaggerated the effects of IL-1β through the upregulation of *IL-1R1* expression (Fig. 8d). This phenomenon was more prominent in BMMs isolated from matured adult mice, as IL-1β, TNF-α, and IFN-γ together led

to an around 20-fold increase in the expression of *IL-1R1*, while IL-1β alone only upregulated *IL-1R1* by fivefold. As a result, two major marker genes (i.e., *MMP9* and *CTSK*) for osteoclastic activities were found to be further upregulated when the three inflammatory cytokines were administered in combination (Fig. 8d). In BMMs isolated from young mice, the addition of TNF-α and IFN-γ to IL-1β resulted in a fourfold increase in *MMP9* expression, when IL-1β alone failed to induce higher *MMP9* expression than the control. Meanwhile, in BMMs isolated from matured adult mice, the combination of IL-1β, TNF-α, and IFN-γ led to a more than sevenfold increase in the expression of *MMP9* and a 1.5-fold increase in the expression of *CTSK*, while IL-1β alone did not significantly alter the expression of these two osteoclastic marker genes. Notably, the three inflammatory cytokines synergistically contributed to a more than 60-fold increase in the expression of Nitric Oxide Synthase 2 (*NOS2*), when IL-1β alone only upregulated *NOS2* expression by less than 30 times. Together, these findings indicated that the pro-inflammatory cytokines synergistically contribute to pathological bone resorption.

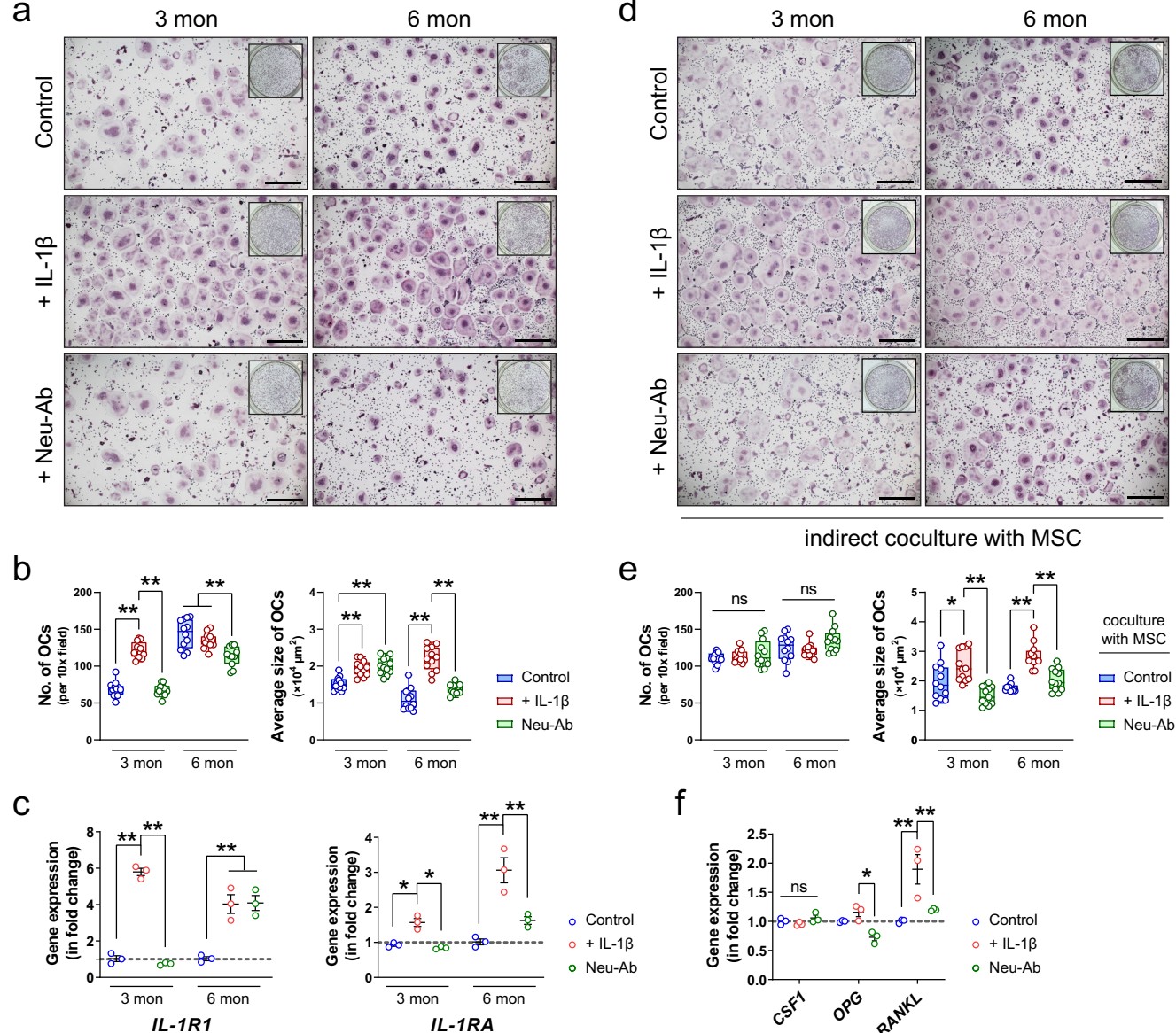

**Fig. 7 Inflammatory cytokine IL-1β promotes osteoclastogenesis. a** Representative microscopic images (scale bars = 500 μm) and **b** the corresponding quantifications (n = 12) showing the formation of TRAP[+] multinuclear cells from bone marrow macrophage (BMM) isolated from young mice (3 months) and matured adult mice (6 months). Recombinant murine IL-1β or its neutralizing antibody (Neu-Ab) was added to the culture medium throughout the osteoclastic induction using M-CSF and RANKL. **c** The gene expression of *IL-1R1* and *IL-1RA* in BMMs from young or matured adult mice with or without the presence of recombinant murine IL-1β or its neutralizing antibody (n = 3). **d** Representative microscopic images (scale bars = 500 μm) and **e** the correspond quantifications (n = 12) showing the formation of TRAP[+] multinuclear cells when the BMMs were indirectly co-cultured with mesenchymal stem cells (MSCs) stimulated with recombinant murine IL-1β or its neutralizing antibody. **f** The gene expression of *CSF1*, *OPG*, and *RANKL* in MSC-treated with recombinant murine IL-1β or its neutralizing antibody (n = 3). Data were mean ± SD. ns: $P > 0.05$, *$P < 0.05$, **$P < 0.01$ by two-way ANOVA with Bonferroni's post hoc test (**b**, **c**, **e**) or one-way ANOVA with Tukey's post hoc test (**f**).

## Discussion

In addition to respiratory tract manifestations, extrapulmonary manifestations are also commonly reported in severe coronavirus infections such as COVID-19, SARS, and MERS[4,26,27]. Based on the phylogenetic similarities of SARS-CoV-2 and SARS-CoV, it has been postulated that the two beta coronaviruses may cause similar clinical features in infected patients. However, recent evidence has increasingly shown that there are more musculoskeletal sequelae associated with COVID-19 than SARS[10,28]. The most severe musculoskeletal complication in SARS patients was non-progressive avascular necrosis of the femoral head caused by high-dose steroid pulse therapy[29]. In contrast,

musculoskeletal sequelae have been increasingly reported in COVID-19 patients including those who have recovered from the acute phase of the infection[10,28]. During the revision of this work, it was also reported elsewhere that SARS-CoV-2 infection can induce bone loss in a lethal human ACE2-transgenic mouse model[30–32]. Nevertheless, it would be important to investigate the long-term bone changes not only in severe COVID-19 cases but also in mild to moderate ones, because most patients recover from the severe inflammation after proper treatment. Thus, using our non-lethal hamster model represent mild to moderate human disease, we showed significant bone resorption happens at the acute inflammatory stage after SARS-CoV-2 infection. Moreover,

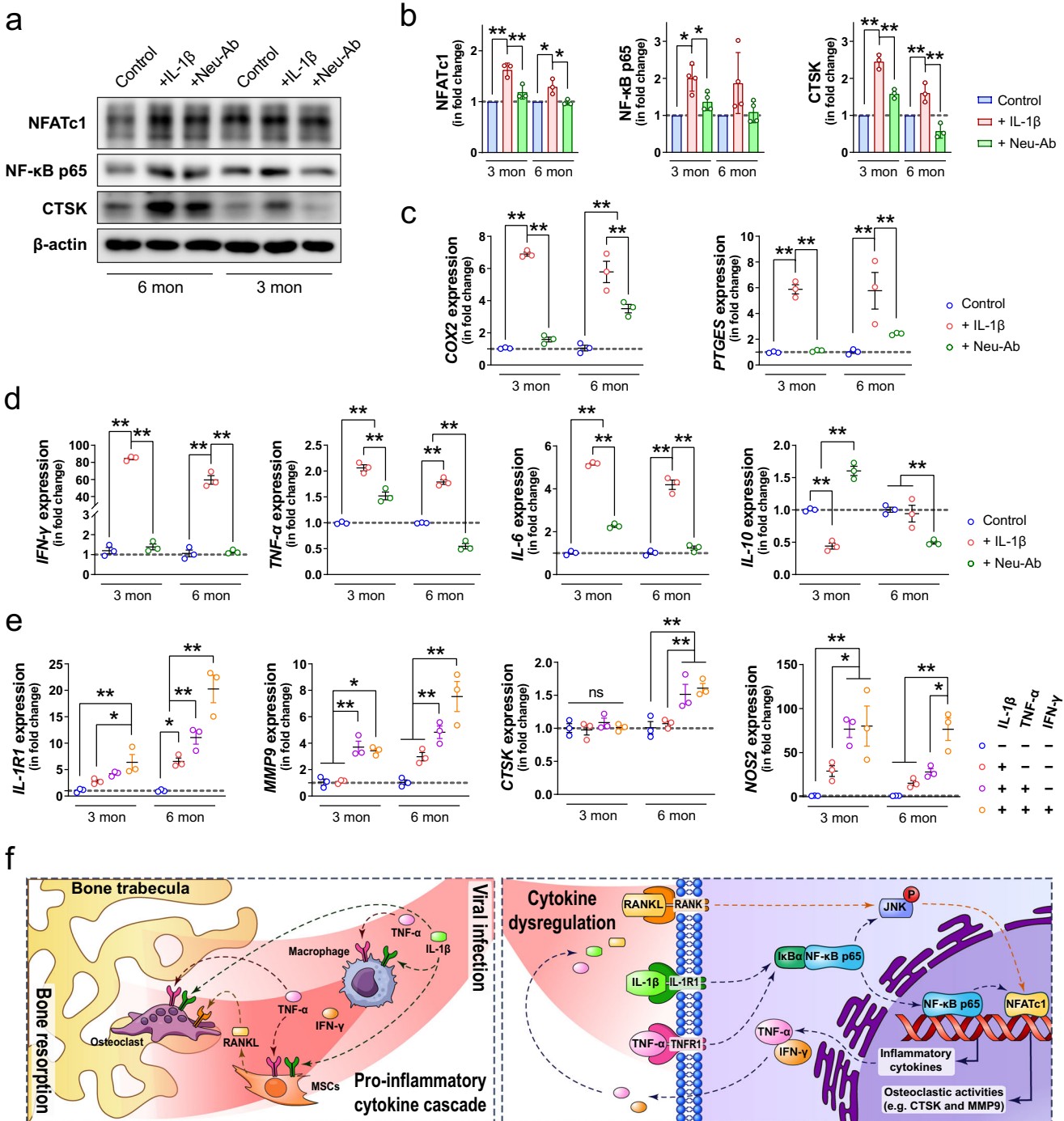

**Fig. 8 Inflammatory cytokines exacerbate the pro-inflammatory response in bone marrow macrophages. a**, **b** Representative Western blots (**a**) and corresponding quantification (**b**) showing the expression of NFATc1 ($n = 3$), NF-κB p65 ($n = 4$), and CTSK ($n = 3$) in BMMs isolated from matured adults (6 months) or young (3 months) mice with or without the presence of recombinant murine IL-1β or its neutralizing antibody. **c**, **d** The gene expression of pain-associated cytokines (**c**) and other pro/anti-inflammatory cytokines (**d**) in BMMs from young or matured adult mice with or without the addition of recombinant murine IL-1β or its neutralizing antibody ($n = 3$). **e** The effects of various inflammatory cytokines on the gene expression of *IL-1R1*, *MMP9*, *CTSK*, and *NOS2* ($n = 3$). **f** Schematic diagram showing SARS-CoV-2 infection-induced cytokine dysregulation leads to an amplified and broadened pro-inflammatory cascade in the skeletal tissue, resulting in pathological bone resorption. Data were mean ± SD. ns: $P > 0.05$, *$P < 0.05$, **$P < 0.01$ by two-way ANOVA with Bonferroni's post hoc test.

in addition to a significantly increased number of RANK$^+$ osteoclast precursors at this stage[33], there were also more TRAP$^+$ osteoclasts expressing NFATc1, which is known to serve as a master regulator for terminal differentiation of osteoclasts[34]. This implies that pathological bone destruction may happen quickly after the onset of COVID-19. More importantly, similar findings

were evident in different bone tissues harvested from the hamsters, suggesting that the bone loss is not site-specific but systemic. Without proper intervention, the bone volume and bone mineral density were barely restored even after the viral load became undetectable at the post-recovery/chronic inflammatory phase. The presence of pathological bone loss may in turn

complicate the rehabilitation of COVID-19 patients. For example, low bone mineral density is a known risk factor for vertebral fractures that may impair the respiratory function of COVID-19 patients in the rehabilitation phase[35,36]. It was recently reported that thoracic vertebral fractures occurred in 36% of COVID-19 patients and increased the patients' need for noninvasive mechanical ventilation[37].

Osteoporosis, which is characterized by a decrease in bone mass, microarchitectural bone disruption, and skeletal fragility leading to a higher fracture rate, has been extensively reported in critically ill patients[38]. The inflammatory cytokines are suggested to be one of the most vital mediators of the pathological bone loss in these diseases because they not only activate osteoclasts but also impede osteoblast function[18]. Cytokine dysregulation has been associated with various clinical manifestations of COVID-19, including some involving the musculoskeletal system, such as myalgia, sarcopenia, arthralgia, and arthritis[39,40]. Our findings in this study demonstrated that SARS-CoV-2-induced pathological bone resorption through a pro-inflammatory cascade instead of direct infection in the skeletal tissue. Indeed, the direct infection of SARS-CoV-2 in bone tissue is very unlikely because there is little to no expression of *ACE2* or co-expression of *TMPRSS2*, which are both vital for viral entry, in the bone marrow[41]. Additionally, a recent study demonstrated that SARS-CoV-2 was not able to bind and infect hematopoietic stem and progenitor cells in bone marrow, from which BMMs were derived[42]. Consistent with our findings, it has been reported that while the viral load in the lung tissue of SARS-CoV-2-infected hamsters dropped from the peak level at 3 dpi to a significantly lower but still detectable level at 9 dpi, the viral load in bone marrow remained at the same level as mock control over the entire study period[43]. Multiplex quantification of immune mediators of SARS-CoV-2 peptide-stimulated cells from different organs also indicated that SARS-CoV-2–specific immune responses were primarily located in the lung or lung-associated lymph nodes instead of the bone marrow[44]. Although we measured a significant increase in serum level of IL-1β, the gene expression of *IL-1β* in bone tissue did not significantly change. This might indicate that the increase in IL-1β level in the bone tissue may be primarily attributed to the SARS-CoV-2-induced immune response in the respiratory system. After the inflammatory cytokines, (e.g., IL-1β and TNF-α) produced in the respiratory tract reached skeletal tissue via circulation, they quickly modulated the monocyte-macrophage lineage residing there to initiate additional pro-inflammatory cytokine cascade in the bone tissue. The cytokines produced in the skeletal system, including but not limited to IFN-γ, IL-6, and PGE2, not only contributed to osteoclastogenesis in several interdependent signaling pathways[18] but also augmented the pro-osteoclastogenic actions of IL-1β and TNF-α by upregulating the expression of IL-1R1, which was implicated to primarily mediate pathological bone resorption[45].

Among the various inflammatory factors that may be associated with osteoclastogenesis[22], we identified IL-1β and TNF-α as the key mediators for SARS-CoV-2-induced bone loss in the hamster model. As two of the most significantly upregulated inflammatory cytokines in the serum samples of SARS-CoV-2-infected hamsters, IL-1β and TNF-α have been known as potent stimulators of bone resorption from as early as the 1980s[46,47]. They are both shown to be essential in RANKL-induced osteoclast formation through the activation of osteoclastogenesis-related signaling pathways such as the c-Jun N-terminal Kinase (JNK) signaling[48,49]. Besides their interdependent roles in mediating inflammatory osteopenia[50], IL-1β and TNF-α were also found to synergistically interact with other pro-inflammatory cytokines to stimulate osteoclastic differentiation[46,51,52]. In this study, we found that the concurrence of three pro-inflammatory cytokines (i.e., IL-1β, TNF-α, and IFN-γ) in the bone tissue after SARS-CoV-2 infection is preferably upregulated *IL-1R1*, which is primarily expressed in pathologically activated osteoclasts responsible for inflammatory bone destruction[53,54]. In contrast, the expression of *IL-1R2*, which serves as a decoy receptor for IL-1β to negatively regulate IL-1β signaling, was not significantly altered. Consequently, these SARS-CoV-2 infection-induced pro-inflammatory cytokines dramatically upregulated the expression of *MMP9* and *CTSK*, which are both known to play dominant roles in the degradation of extracellular matrix[55]. Additionally, we also found that the inflammatory cytokines elevated in COVID-19 (e.g., IL-1β) were able to promote the formation of osteoclasts via regulating the production of RANKL from MSCs, as reported elsewhere[56].

As we have demonstrated the SARS-CoV-2-induced bone loss in adult hamster, we further involved in vitro experiments using BMMs isolated from 3-month-old (young adult) or 6-month-old (matured adult) mice to test whether the inflammatory-mediated osteoclastogenesis is age-related. Importantly, we demonstrated that BMMs isolated from young adult mice were more responsive to the pro-osteoclastic stimulation of IL-1β. This is clinically relevant because young patients generally have a stronger ability to adequately respond to viral infections with rapid production of a high level of pro-inflammatory cytokines[57,58]. This heightened pro-inflammatory response, together with the lower baseline pro-inflammatory state in young patients, makes them more susceptible to various syndromes related to immune dysregulation[58,59]. Meanwhile, we also found that the involvement of other pro-inflammatory cytokines, such as TNF-α and IFN-γ, contributed to a more prominent effect on promoting the osteoclastic activities in BMMs isolated from matured adult mice than the ones from young mice. This might explain why musculoskeletal symptoms are mostly seen in adult patients rather than children and the elderly[60]. Additionally, we noticed that the neutralizing antibody failed to block the late osteoclastic differentiation, especially in BMMs from young mice, as the size of osteoclasts and the expression of CTSK did not drop to the baseline level after the addition of the IL-1β neutralizing antibody. These data suggest that multiple neutralizing antibodies targeting different pro-inflammatory cytokines may need to be considered for limiting IL-1β-induced osteoclastogenesis. As demonstrated in our study that the initial stimulation of IL-1β led to an amplified and broadened pro-inflammatory cascade, resulting in the production of several other cytokines favoring osteoclastogenesis, simultaneous block of IL-1β and other pro-inflammatory cytokines would be required for preventing bone loss[61].

Besides the immunomodulatory effect and the pro-osteoclastogenesis effect, the accumulation of various pro-inflammatory cytokines in the skeletal tissue can lead to several other long-term health concerns. For example, we showed that IL-1β dramatically upregulated *COX2* and *PTGES*, which both contribute to the production of PGE2[62]. PGE2 is not only an inflammatory mediator involved in bone modeling but also a neuromodulator that can sensitize peripheral sensory neurons leading to inflammatory pain[63]. More than 20% of COVID-19 patients report lasting (from week 0 to week 28) bone ache or burning feeling[10]. Therefore, long-term monitoring of the inflammatory status of bone tissue after the recovery of the disease would be necessary. As a major indicator but not the sole driver in the pathology of COVID-19[64], IL-6 is commonly found significantly increased in patients with exacerbating disease progression[64]. However, many studies have shown that the serum level of IL-6 would not be significantly upregulated in hamsters after SARS-CoV-2 infection[65,66], because SARS-CoV-2 would not cause severe or even lethal disease in hamsters. Instead, the virus

would induce a mild to moderate pathologic procedure resembling the clinical situation found in most COVID-19 patients. Therefore, our observation suggests that the inflammatory bone loss can happen in mild to moderate infection or post-recovery cases of COVID-19 patients. The inflammation in bone tissue can also alter the output of immune cells or cytokines from the bone marrow, which are supposed to participate in combating viral infection[67]. In our study, the significantly upregulated expression of *IFN-γ* and its signaling-associated genes (e.g., *IRF1* and *IRF2*) in bone tissue indicate they might play an essential role in protecting bone tissue from virus infection[68]. However, it is also noteworthy that the expression of several anti-viral chemokines and cytokines, such as IFN-β[69], IL-21[70], and CCL17[71], remained unchanged or even downregulated in the bone tissue. Thus, the suppression of the bone marrow-derived anti-viral factors may be one of the immune-evading mechanisms for SARS-CoV-2 that warrants further investigation.

In a recently published meta-analysis, Anakinra, a recombinant IL-1 receptor antagonist, was suggested as a safe, anti-inflammatory treatment option to reduce the mortality risk in patients admitted to hospital with moderate to severe COVID-19 pneumonia, especially in the presence of signs of hyperinflammation[72]. This suggests that adequate use of an anti-inflammatory treatment may be beneficial for COVID-19 during both the acute phase and the chronic phase involving long-term inflammatory complications. Therefore, evaluation of the effects of anti-inflammatory agents such as IL-1 receptor antagonists and IL-1 neutralizing antibodies on the prevention and/or treatment of SARS-CoV-2 infection-induced inflammatory bone loss should be included in future studies. In addition, there are several limitations to our study. First, due to the lack of commercialized detection assays for evaluating biomolecules of hamster at the protein level, we only tested the circulating levels of IL-1β, TNF-α, and IL-6, which are known as three of the most important pro-inflammatory cytokines of the innate immune response. Nevertheless, the striking similarities in the cytokine dysregulation between hamster model and human patients have been verified elsewhere using single-cell RNA sequencing[73]. Secondly, given the fact that the current hamster model mainly represents mild to moderate disease in human, additional investigations on the characteristics and treatment options for SARS-CoV-2-induced bone loss should be further conducted in more severe and/or lethal models.

In this study, we demonstrated the influence of SARS-CoV-2 infection on systemic bone loss during the acute and post-recovery/chronic phases. We revealed the pro-inflammatory cytokines derived from the respiratory system as the major mediators for pathological bone resorption. These pro-inflammatory cytokines disrupt the balance in bone metabolism and trigger another pro-inflammatory cascade in the skeletal tissue to further augment their pro-osteoclastogenesis effect (Fig. 8e). The findings in our study highlight the need to closely monitor COVID-19 patients' bone density. The benefits of prophylactic or therapeutic interventions against the development of pathological bone loss in COVID-19 patients should be further evaluated in animal models and clinical trials.

## Methods

**Virus and biosafety**. SARS-CoV-2 (strain HKU-001a, GenBank accession number: MT230904) was a clinical strain isolated from the nasopharyngeal aspirate specimen of a COVID-19 patient in Hong Kong[74]. The plaque-purified viral isolate was amplified by one additional passage in VeroE6 cells to make working stocks of the virus as described previously[75]. All experiments involving live SARS-CoV-2 followed the approved standard operating procedures of The University of Hong Kong (HKU) Biosafety Level-3 facility[76,77].

**Animal model**. The animal experiments were approved by the HKU Committee on the Use of Live Animals in Teaching and Research. Briefly, 6–10-week-old male or female golden Syrian hamsters (*Mesocricetus auratus*) were obtained from the Chinese University of Hong Kong Laboratory Animal Service Center through the HKU Center for Comparative Medicine Research. The animals were kept in Biosafety Level-2 housing and given access to standard pellet feed and water ad libitum until a virus challenge in our Biosafety Level-3 animal facility. Each animal was intranasally treated with $10^5$ PFU of SARS-CoV-2 in 50 μL of PBS under intraperitoneal ketamine (200 mg/kg) and xylazine (10 mg/kg) anesthesia at 0 dpi as we previously described[78]. Mock-infected animals were treated with 50 μL of PBS. Their blood, bone, and lung tissues were collected at sacrifice at 4, 30, and/or 60 dpi for μCT, virological, and histopathological analyses.

**Micro-CT analysis**. The PBS or virus-treated hamsters were sacrificed at 4, 30, and 60 dpi for micro-CT analysis following a standard guideline described previously[79]. Before being transferred from the biosafety Level-3 facility, the specimens were fixed in 4% paraformaldehyde for 48 h and 70% ethanol for 24 h to inactivate the pathogens. The bone specimens were scanned by a high-resolution micro-CT scanner (SkyScan 1076, Kontich, Belgium) at a resolution of 8.665 μm per pixel. The voltage of the scanning procedure was 88 kV with a 110-μA current. Two phantom-contained rods with a standard density of 0.25 and 0.75 g/cm³ were used for calibration of bone mineral density (BMD). Data reconstruction was done using the NRecon software (Skyscan Company), the image analysis was done using CTAn software (Skyscan Company), and the 3D model visualization was done using CTvox (Skyscan Company) and CTvol (Skyscan Company). The region of interest (ROI) contained 200 layers of images beginning from the distal metaphyseal growth plate of femurs or the proximal metaphysis growth plate of tibias (Fig. 1b). Trabecular bone parameters, including bone volume fraction (BV/TV), specific bone surface (BS/BV), bone mineral density (BMD of TV), trabecular thickness (Tb.Th), trabecular number (Tb.N), trabecular pattern factor (Tb.Pf), trabecular separation (Tb.Sp), polar moment of inertia (ρMOI), and structure model index (SMI), as well as cortical bone parameters, including cortical bone area (Ct.Ar) and cortical bone thickness (Ct.Th) were measured from the μCT data.

**Histological analysis**. Histology and immunohistochemical staining were performed on both paraffin sections and cryosections. In brief, the bone specimens, after fixation in 4% PFA for 48 h, were decalcified with 12.5% ethylenediaminetetraacetic acid (EDTA, Sigma-Aldrich) for 4 weeks. For paraffin sections, the specimens were processed, embedded in paraffin, and cut into 5-μm-thick sections using a rotary microtome (RM215, Leica Microsystems, Germany). Haematoxylin and eosin (H&E) staining, TRAP staining (Sigma-Aldrich), and ALP staining (Sigma-Aldrich) were performed on selected sections from each sample following the manufacturer's instructions. Images were captured using the Vectra Polaris Imaging System (Akoya Biosciences, USA). For immunostaining, the samples were dehydrated in 20% sucrose solution with 2% polyvinylpyrrolidone (PVP, Sigma-Aldrich) for 24 h and embedded in 8% gelatin (Sigma-Aldrich) supplemented with 20% sucrose and 2% PVP. Forty-μm-thick coronal-oriented sections of the femurs were obtained using a cryostat microtome. Immunostaining was performed using a standard protocol. Briefly, after blocking with 10% goat serum, the sections were incubated with primary antibodies to CD68 (Abcam, ab31630/ab125212), RANK (Abcam, ab13918), TRAP (Abcam, ab216025), osteocalcin (TAKARA, M186), IL-1β (Abcam, ab9722), IL-1RA (Abcam, ab124962), TNF-α (Abcam, ab9635), IFN-γ (Abcam, ab9657), NF-κB (CST, #8242), anti-NFATc1 (CST, #8032) overnight at 4 °C. Alexa-Fluor 488-conjugated and Alexa-Fluor 647-conjugated secondary antibodies (Thermo Fisher Scientific) were used for immunofluorescent staining, while the nuclei were counterstained with Hoechst 33342 (Thermo Fisher Scientific). Immunofluorescent images were captured using LSM 780 confocal microscopy (Zeiss, Germany).

**Multiplex IHC analysis**. Antigen retrieval and blocking were performed on the dewaxed slides using the Antigen retrieval reagent (pH 6.0) and Blocking/antibody diluent provided in the Opal Polaris Multicolor Manual IHC Detection Kit (Akoya Biosciences, USA), following the manufacturer's instructions. In brief, the incubation of each primary antibody was done overnight at 4 °C. The primary antibodies used in this study included anti-CD68 (Abcam, ab31630, USA), anti-TRAP (Abcam, ab216025), anti-RANK (Abcam, ab13918), anti-IL-1β (Abcam, ab9722), anti-NP (Thermo Fisher, USA), and anti-ACE2 (Thermo Fisher). Between each incubation of the primary antibody, tyramide signal amplification (TSA) visualization was performed using the Opal Polymer Horseradish peroxidase (HRP) secondary antibody and fluorophores: Opal 520, Opal 570, Opal 620, Opal 690, and DAPI (Akoya Biosciences, USA). The stained slides were imaged using the Vectra Polaris Imaging System (Akoya Biosciences, USA).

**Cell culture**. The mesenchymal stem cells (MSCs) and bone marrow macrophages (BMMs) were isolated from the long bones of 3-month-old or 6-month-old C57L6/J mice. In brief, the mice were euthanized with overdosage of intraperitoneal injection of Pentobarbital. After the removal of attached soft tissues using forceps and gauze, the dislocated femurs and tibias were dissected into pieces. The whole bone marrow cells were resuspended in a serum-free DMEM medium (Gibco, USA) using a vortex mixer, while the bone chips and debridement were removed by passing the mixture through a cell strainer. After centrifugation, the cell pellet

was resuspended in a DMEM medium, supplemented with 10% FBS and 1% Penicillin-Streptomycin (complete DMEM medium), and cultured in culture flasks. After a 6-h incubation in a humidified incubator with 5% $CO_2$ at 37 °C, the unattached cells were gently removed for the induction of BMMs. Meanwhile, the attached bone marrow cells were gently washed in PBS and further cultured as MSCs in a complete DMEM medium until they reached 80% confluence. The culture medium was refreshed every 2 days, and only passages 3–5 were used for the experiments. After the 3-day macrophage induction using a complete DMEM medium supplemented with 20 ng/ml of macrophage colony-stimulating factor (M-CSF), the BMMs became adherent for the osteoclastic differentiation using a complete DMEM supplemented with 50 ng/mL of receptor activator of nuclear factor-kappa-B ligand (RANKL, R&D System, USA) and 20 ng/mL of M-CSF (R&D System, USA). The coculture of BMMs with MSCs was done using a transwell system (0.4 µm pore size, Corning Costar, USA). In brief, MSCs were seeded in the transwell inserts and cultured overnight with 5% $CO_2$ at 37 °C for attachment. Afterward, the transwell inserts were placed into the lower chambers in which BMMs have been differentiated using a medium supplemented with M-CSF for three days as described previously. To mimic the challenge of the pro-inflammatory microenvironment in COVID-19, 1 ng/mL recombinant murine IL-1β (R&D system, USA) was supplemented in the medium. For the inhibition of IL-1β, 10 ng/mL IL-1β neutralizing antibody (Abcam, ab9722) was added to the cell culture. For studying the synergistic effects of pro-inflammatory cytokines on BMMs, the culture medium of BMMs was further supplemented with 1 ng/mL TNF-α (R&D system, USA) and 1 ng/mL IFN-γ (R&D system, USA).

**Real-time quantitative polymerase chain reaction (RT-qPCR) assay**. The total RNA from the bone specimens were isolated using Trizol reagent (Thermo Fisher, USA) following the manufacturer's instructions. The total RNA from the cultured cells was extracted and purified using an RNeasy Plus kit (Qiagen, USA) following the manufacturer's instructions. For the reverse transcript, complementary DNA was synthesized using a Takara RT Master Mix (Takara, Japan), following the manufacturer's instructions. The primers used in the RT-qPCR assay were synthesized using Integrated DNA Technologies (IDT, Singapore), based on sequences designed using Primer-BLAST (National Center for Biotechnology Information, NCBI, Supplementary table 1) or retrieved from the Primer Bank (http://pga.mgh.harvard.edu/primerbank/, Supplementary table 2). The SYBR Green Premix Ex Taq (Takara, Japan) was used for the amplification and detection of cDNA targets on the LightCycler480 Real-time PCR system (Roche, USA). The mean cycle threshold (Ct) value of each target gene was normalized to the housekeeping genes (i.e., RPL18 or GAPDH). The results were shown in a fold change using the ΔΔCt method.

**Western blotting**. The protein from homogenized bone tissue from hamster were isolated using Trizol reagent (Thermo Fisher, USA) following the manufacturer's instructions. The protein from cultured BMM was harvested using RIPA Lysis and Extraction Buffer (Thermo Fisher, USA) containing a 1% Phosphatase Inhibitor Cocktail (Thermo Fisher, USA). The concentration of protein was measured using the BCA Protein Assay Kit (Thermo Fisher, USA). A total of 20 µg of protein from each sample was subjected to sodium dodecyl sulfate-polyacrylamide gel electrophoresis (SDS-PAGE) and transferred to the polyvinylidene difluoride (PVDF) membrane (Merck Millipore, USA). Then, the membrane was blocked in 5% w/v bovine serum albumin (BSA, Sigma-Aldrich, USA) and incubated with blocking buffer-diluted primary antibodies overnight at 4 °C. The primary antibodies used include mouse anti-NFATc1 (Santa Cruz, USA), mouse anti-TRAP (Abcam), rabbit anti-Cathepsin K (Abcam), mouse anti-RANK (Abcam), rabbit anti-NF-κB p65 (CST), rabbit anti-IL-1β (Abcam), rabbit anti-IL-1RA (Abcam), rabbit anti-TNF-α (Abcam), rabbit anti-phospho-JNK (CST), rabbit anti-JNK (CST), rabbit anti-β-actin (Abcam). The protein bands were visualized by using HRP conjugated secondary antibodies and an enhanced chemiluminescence (ECL) substrate (Advansta, USA) and exposed under the Typhoon5 Biomolecular Imager 680 (GE Amersham, USA).

**ELISA assay**. The serum samples of the hamsters were collected at 4 dpi or 30 dpi for cytokine/chemokine analysis. The serum level of IL-1β, TNF-α, and IL-6 were detected using a specific ELISA kit (MyBiosource, USA) following the manufacturer's instructions.

**Statistical analysis**. All data analyses were performed and illustrated using the Prism software (version 7, GraphPad, USA). The results were expressed as means ± standard deviation (SD). The exact sample size (n) for each experimental group was clearly shown as dot plots in the figures and indicated in the legends. For comparisons among multiple groups, a one-way or two-way analysis of variance (ANOVA) was used, followed by Tukey's multiple-comparison post hoc test or Bonferroni's post hoc test. The levels of significant difference among the groups were defined and noted as $*P < 0.05$ and $**P < 0.01$. The sample size was decided based on preliminary data, as well as on observed effect sizes.

## Data availability

All relevant data that support the findings of this study are available within the Article and Supplementary Information or from the corresponding author upon reasonable request. Source Data file is provided with this paper.

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

## Acknowledgements

We sincerely thank the staff members at the Faculty Core Facility, Li Ka Shing Faculty of Medicine, The University of Hong Kong, the Centre for Comparative Medicine Research of The University of Hong Kong, and the Laboratory Animal Service Centre of The Chinese University of Hong Kong for their facilitation of the study. K.W.-K.Y. received funding from the National Key R&D Program of China (2018YFA0703100), Health and Medical Research Fund (19180712), Hong Kong Innovation Technology Fund (ITS/287/17 and ITS/405/18), Science and Technology Commission of Shanghai Municipality (18410760600), International Partnership Program of Chinese Academy of Sciences (GJHZ1850). K.W.-K.Y. and K.M.-C. C. received funding from General Research Fund (17207719, 17214516, and

N_HKU725/16), HKU-SZH Fund for Shenzhen Key Medical Discipline (SZXK2020084), Shenzhen Science and Technology Funding (JSGG20180507183242702), Sanming Project of Medicine in Shenzhen, China (SZSM201612055), Guangdong Financial Fund for High-Caliber Hospital Construction (174-2018-XMZC-0001-03-2125/D-10). J.F.-W.C. received funding from Collaborative Research Fund (C7060-21GF) of the Research Grants Council, Hong Kong Special Administrative Region, the National Key R&D Program of China (2020YFA0707500 and 2020YFA0707504), Health and Medical Research Fund (20190572 and COVID190121), the University of Hong Kong Outstanding Young Researcher award, the University of Hong Kong Research Output Prize, and donations from Lo Ying Shek Chi Wai Foundation. K.-Y.Y. and J.F.-W.C. received funding from the Theme-Based Research Scheme (T11-709/21-N) of the Research Grants Council of HKSAR, Sanming Project of Medicine in Shenzhen, China (SZSM201911014), the High Level-Hospital Program of the Health Commission of Guangdong Province, Health@InnoHK of Innovation and Technology Commission, the Government of Hong Kong Special Administrative Region, the Consultancy Service for Enhancing Laboratory Surveillance of Emerging Infectious Diseases and Research Capability on Antimicrobial Resistance for Department of Health of the HKSAR Government, the Major Science and Technology Program of Hainan Province (ZDKJ202003), the Hainan Talent Development Project (SRC200003), the research project of Hainan Academician Innovation Platform (YSPTZX202004), the Collaborative Project (EKPG22-01) of Guangzhou Laboratory, the Emergency COVID-19 Project (2021YFC0866100), Major Projects on Public Security, National Key Research and Development Program. K.-Y.Y. and J.F.-W.C. are also grateful for the donations of May Tam Mak Mei Yin, Richard Yu, and Carol Yu, the Shaw Foundation of Hong Kong, Michael Seak-Kan Tong, the Providence Foundation Limited (in memory of the late Lui Hac Minh); Lee Wan Keung Charity Foundation Limited, Hui Ming, Hui Hoy, and Chow Sin Lan Charity Fund Limited, Hong Kong Sanatorium & Hospital, Chan Yin Chuen Memorial Charitable Foundation, Tse Kam Ming Laurence, Marina Man-Wai Lee, the Hong Kong Hainan Commercial Association South China Microbiology Research Fund, the Jessie & George Ho Charitable Foundation, Perfect Shape Medical Limited, Kai Chong Tong, Foo Oi Foundation Limited, Betty Hing-Chu Lee, and Ping Cham So. The funding sources had no role in the study design, data collection, analysis, interpretation, or writing of the report.

## Author contributions

W.Q. performed the in vitro and in vivo tests, as well as analyzed and interpreted the data. H.E.L. and H.X. contributed to the in vitro tests and helped with the data analysis. V.K.-M.P. and C.C.-S.C. contributed to the infection of hamsters and the collection of specimens. H.C., S.Y., T.T.-T.Y., K.K.-H.C., J.O.-L.T., C.C.-Y.C., J.-P.C., and C.L. performed experiments and analyzed data. K.-Y.Y. and K.M.-C.C. provided funding and interpreted data. J.F.-W.C. and K.W.-K.Y. contributed to the study design, data interpretation, and supervision of the project. W.Q., J.F.-W.C., and K.W.-K.Y. wrote the manuscript, with input from all authors.

## Competing interests

The authors declare no competing interests.
