## [Peer Review File · Nature Communications]

Reviewers' Comments:

Reviewer #1:

Remarks to the Author:

This is a very interesting and timely report which corroborates recently published and presented findings by at least 2 other groups, but extends the work by examining a different animal model (hamster vs mouse), looks at multiple bone sites, looks at multiple times post-infection, and delves into some additional correlative findings (some statements about mechanism are a reach). The authors should be congratulated on trying to tackle an important problem. The study could be improved with modifications in several areas as detailed below. They miss a key opportunity to understand whether the cytokine levels/expression change over time, perhaps a drop in the 30 or 60 dpi would signify bone loss may be able to recover but it takes time. Alternatively, if levels remain high 60 dpi, even greater loss may be possible over time. Discussing these possibilities along with the higher likelihood of these findings translating into humans as these authors are now confirming the findings of others in a mouse model of COVID are important elements that should be flushed out.

Major Concerns

1. The authors need to revise the manuscript to reflect the recent publication and published abstracts presented at the American Society for Bone and Mineral Research in early October 2021. These show that others have already published/presented that bone loss occurs in mouse models of COVID. Thus, their data corroborate those studies and extend these observations. Further, since this is shown in another animal, it adds to the idea that this observation may in fact translate into humans. This should be emphasized and expanded upon. (PMID: 34607050 AND abstract #1025: Blaine Christian presenting author and abstract #LB-1105: Christopher Dalloul presenting author).
2. The established guidelines in the field for uCT assessments and reporting should be followed (PMID: 20533309). Several of the abbreviations are wrong (Tb.Th – is trabecular thickness) and the methods is not complete and should be expanded for others to reproduce results. SMI and Tb.Sp should be reported as these would be valuable parameters to understand.
3. Additional expansion in the discussion on how to reconcile differences in your findings from the literature is required. For example, abstract #1025 shows virus in bone tissue and bone marrow macrophages have also been shown to have virus within them. These references should be added and discussed in the context of the authors findings.
<https://www.biorxiv.org/content/10.1101/2021.04.14.439793v2.abstract>
4. Additional discussion and clarification of differences in circulating, vs bone tissue, vs cell findings of the interleukins (and other cytokines) discussed should be included. As it currently stands, some elements are confusing to understand after multiple readings of these sections.
5. It is unclear where all of the samples went or why some figures have different time points etc. For example why is a 60 day mock not included in Figure 1A? Were those not done? Which mock dosing is shown in the figures – it is not clear if 4dpi, 30dpi or what? It would be better to show all of the mock dosing or compare for each time point. That is the proper control for 60 dpi infected is to have a 60 dpi mock-infected. Related to this why was a different volume of vehicle used for the infection? It is known that mock-infection can also cause pneumonia and other lung issues and that needs to be properly accounted for and volume of liquid the mice are inhaling can impact this. These limitations minimally need to be addressed and acknowledged. Likewise, some figures show fewer samples than what appears to be listed in the methods, these differences should be explained.
6. In figure 2, why is 4 dpi not shown in panel C, why is 60 dpi not shown in panels d & e, addition of these in the figure and similarly showing these data in the subsequent figures are important to understand the whole picture. For example, in Figure 3 & 4, having the data would demonstrate whether the levels are only up early (during the acute phase) and then decline during the recover phase. This key understanding would significantly improve our understanding of what is occurring with SARS-CoV-2 infection and mechanistically is more important than repeating wild-type osteoclast studies with treatments of cytokines/antibodies. The authors should have these samples

based on their study design and other data presented.

7. Many of the mouse cell osteoclast studies which the authors are considering the mechanism of how the cytokine storm is impacting bone loss in their model are studies that others have done over the years and should be appropriately acknowledged.

8. Your graphical abstract and Figure 8 panel e need to be modified. Why would IL-6 be included in the figure? Your data show that IL-6 serum levels are no different between mock and 4 dpi animals. Your bone RT-PCR data show that IL-6 is down in the bones of 4dpi animals compared to mock-infected animals. So how can your figure say that is part of the pathway? Your mouse cell data suggests it is up but your in vivo infection data does not show that. This contradiction needs to be flushed out in the discussion. And text related to the role of IL-6 needs to be modified accordingly.

9. Although I agree 3 month mice are younger than 6 month mice, both are considered adults but 6 months is NOT considered middle aged. What was the rationale for using 2 ages of the cells in Figure 7&8, and now how does one explain the differences. More discussion about the differences over time is required for these data as it is difficult to reconcile.

10. Discussion should include future directions which would according to the focus of the examination of IL-1B in wild-type cells be to perform in vivo infection studies and treat animals with neutralizing antibody and see if bone parameters improve and osteoclast numbers are reduced. The data presented are all hypothetical and correlative, not truly mechanistic as in the infection model no interventions were provided to test whether the cytokine storm is truly responsible. There are places throughout the manuscript where the interpretation is stretched in this regard. Completion of this study and showing an improvement in bone indices is truly the best mechanistic study for this manuscript. Last paragraph of the introduction needs revising as not accurate, same in the abstract and discussion. Carefully only stating what you have shown is critical.

11. I am not sure that the idea of a second wave is accurate. If others have documented this please refer to those data. Second suggests difference over time and I am not clear with the data shown that it is not all from the same insult – systemic cytokine storm, all in the first wave to use your terminology. Clearly, modifying Figure 8e based on whether this is your hypothesis or accurate may be needed.

Minor Concerns.

1. There are some grammatical errors which should be carefully checked during revisions for ease of reading. (line 173 as example)

2. Western, in Western blot should be capitalized throughout (including supplementary materials).

3. Unless published does not allow, all figures should discuss age and sex of hamsters so reader isn't left waiting for methods to understand – or at least write early in results. In fact Figure 1A could have this information and how many hamsters are in each group to make the study design more clear.

4. Figure 5 panel A and b should be revised for clarity – panel A should have label on rows with Mock and 4dpi. Fig 5b should be all red squares since all 4 dpi (I presume, but this needs clarification).

Reviewer #2:

Remarks to the Author:

In this study entitled "SARS-CoV-2 infection induces inflammatory bone loss in golden Syrian hamsters", the authors reported the novel findings of pathological bone loss after SARS-CoV-2 infection in the Syrian hamster model. They showed that SARS-CoV-2 infection consistently causes significant bone loss of bone trabeculae in both the long bones (femur and tibia) and lumbar vertebrae of the hamsters. They serially studied the bone losses and showed that the changes

progressively worsened from the acute phase at 4 days post-infection to the post-recovery phase at 30-60 dpi. The authors then investigated the underlying mechanism of these SARS-CoV-2-induced bone losses and showed that virus-induced cytokine dysregulation upregulated osteoclastic differentiation of monocyte-macrophage lineage; whereas the osteoblastic differentiation was not significantly altered.

The findings of this work is novel, important, and timely especially as “long COVID-19” or “chronic COVID-19” complications are increasingly being reported in patients who have recovered from the initial infection. The team in this work has excellent track records in coronavirus / COVID-19 research and is the group that reported the first established Syrian hamster model for COVID-19 (Chan JF et al., Clin Infect Dis 2020). The methods and results are well described and the figures are of high quality to illustrate the findings. The authors’ proposed reasons why these bone changes may have been overlooked in patients (in the Introduction) are reasonable and clinically relevant. Coincidentally, the Reviewer noted that a very recent paper describing bone losses in mice was published on 2nd October 2021 (Awosanya OD et al., Bone 2021) after the current work was submitted to Nature Communications. However, the Reviewer finds that the current submission is superior to the published paper for a number of major reasons:

1) The animal model used in the current submission (Syrian hamster) is well-established and closely mimics mild-to-moderate infection in human. Comparatively, the animal model using K18-hACE2 transgenic mice in the Bone paper is a partially to fully lethal model (depending on virus inoculum) and does not suitably represent the majority of patients’ mild-to-moderated COVID-19 clinically. This is especially clinically relevant because long COVID-19 complications occur in surviving patients and not those who have already succumbed during the acute phase.

2) Phenotypically, the current study is more comprehensive which describes the multifocal involvement as well as serial changes in the skeletal system of the infected vs mock-infected hamsters. These data on different bones and longitudinal effects at different time points after SARS-CoV-2 infection were not available in the Bone paper.

3) There was little, if any, mechanistic data provided in the Bone paper (only provided a “hypothetical model” without any experimental data to support the proposed mechanism). Comparatively, the authors in this study used a number of well-designed in vitro and in vivo experiments to demonstrate SARS-CoV-2-induced cytokine storm and the subsequent immune dysregulation in bone microenvironment as the underlying mechanism contributing to the osteoclastogenesis. In particular, this study identified IL-1 β and its receptor IL-1R1 as the key signaling involved in inflammatory SARS-CoV-2-induced bone loss, which may further serve as a potential therapeutic target for COVID-19.

Overall, the findings in this study are important and timely. A number of comments / clarifications listed below should be addressed before further consideration for publication:

Major comments:

1. The decrease in bone density happens during the aging process. Can the difference in bone density caused by the natural aging process? It would be important to clearly show the accurate age of the hamsters involved in this study. Is the age of hamsters in mock control group same as the one in infected group? Please specifically state the ages of the hamsters in the Methods or Results.
2. The study demonstrates significant loss of bone trabeculae, but what about the cortical bone? Is there any change (thickness or density) after the infection of SARS-CoV-2?
3. To ensure the robustness of viral antigen detection, have the authors tried more than one antibody for the detection of viral nucleocapsid protein?
4. It is suggested that the infected hamsters generally recovered at about 7 to 10 dpi, however, it seems the bone loss primarily happen between 4 to 30 dpi, why?
5. Patients with long COVID-19 may have muscle and/or joint pain. Can the authors show data on whether virus infection in this Syrian hamster model?

Minor comments:

1. What do the arrow heads indicate in Figures 4b, 4d, and 5a?
2. If only IL-1 β neutralizing antibody was used, it should be Neu-Ab instead of Neu-Abs.
3. Quantification of WB in Fig. 8a should be provided.
4. In Fig. S3c, it should be NF κ B (should be consistently spelt as NF- κ B).
5. Typos: anti-RANK (Abcam, ab13918) in the Method part.

Reviewer #3:

Remarks to the Author:

Yeung et al. describe inflammatory bone loss in golden Syrian hamster model induced by SARS-CoV-2 infection. They show convincing micro-CT scans in different bones and perform experiments of gene expression at the RNA and protein levels (RT-qPCR, western blot, immunofluorescence, etc) in hamster or tissues or mouse BMMs. They find that there is a first and a second wave of cytokine storm mediated by IL-1B and other cytokines, the second wave being the main responsible for the bone loss. Although the experiments described are elegant and the importance of the findings is high, the paper needs a better discussion of the mechanism proposed – that IL-1beta plays an important role in bones loss through cytokine storm. While the mouse experiments appear to hint at this mechanism, the levels of recombinant IL1-beta in this experiment are likely much higher than the increase observed in hamsters (~25%).

- Do the authors think that an increase in IL1-B of about 25% (Fig 5c) would be able to explain the phenotypes observed? In the experiments of figures 7 and 8, by how many fold was the expression of IL-1beta increased in the presence of recombinant IL-1beta?

- The authors postulate that a cytokine storm leads to bone loss. However, on figure 5c, the increase noted in IL-1-beta and TNF-alpha seem very small to be considered as "cytokine storm". Cytokine storm usually means a large increase in cytokines in a short period of time. Further, only a few cytokines/chemokines are measured, and most are shown as difference in gene expression rather than protein levels. How do the authors support the conclusion of a cytokine storm with so few measurements?

- What is the rationale for using only male hamsters during these studies when there are known sex differences on bone density and health.

Minor comments:

- The authors do immunofluorescence in hamster tissue (such as figure 3E), which can be technically difficult given the relative non-cross reactivity between hamsters and other rodents. Can the authors explain better in the text if they tested many antibodies or optimized a special protocol? This would increase rigor and reproducibility between hamster models of SARS-CoV-2 infection models.

- Authors use the expression "challenged with PBS", but PBS is not a challenge, so this should be corrected.

- I did not see if the authors mention anywhere in the paper: was it previously known that IL1-beta can promote osteoclastogenesis?

- The IL-1B neutralizing antibody sometimes has a similar effect as the IL-1B, such as in panels 8A, 7B and 7C (6 months). Is this expected? The authors should discuss these findings.

- On line 198, the authors say that viral RNA was not detected in bone tissue. But on figure 5b the viral RNA is quantified in bone tissue. Please clarify.

- Figure 5: How can the authors explain the differences in IL-6 on figures 5c (trending to increase but n.s.) and 5e (decreasing)?

- In figure S4, panel c p-JNK does not seem to be changed. In this case, it would be better to say "phosphorylated JNK" instead of "phosphorylation of JNK", for improved clarity.

- On line 420, why do the authors use only 50 microliters of PBS when 100 microliters of SARS-2 were used?

We wanted to thank the reviewers for their thoughtful, constructive and accurate comments for our manuscript. We have addressed all the questions and comments brought forth through additional experimentation and clarification. We have highlighted all changes in the revised manuscript.

REVIEWER COMMENTS:

Reviewer #1 (Remarks to the Author):

This is a very interesting and timely report which corroborates recently published and presented findings by at least 2 other groups, but extends the work by examining a different animal model (hamster vs mouse), looks at multiple bone sites, looks at multiple times post-infection, and delves into some additional correlative findings (some statements about mechanism are a reach). The authors should be congratulated on trying to tackle an important problem. The study could be improved with modifications in several areas as detailed below. They miss a key opportunity to understand whether the cytokine levels/expression change over time, perhaps a drop in the 30 or 60 dpi would signify bone loss may be able to recover but it takes time. Alternatively, if levels remain high 60 dpi, even greater loss may be possible over time. Discussing these possibilities along with the higher likelihood of these findings translating into humans as these authors are now confirming the findings of others in a mouse model of COVID are important elements that should be flushed out.

Response: We thank the reviewer for the encouraging comments.

Major Concerns:

1. The authors need to revise the manuscript to reflect the recent publication and published abstracts presented at the American Society for Bone and Mineral Research in early October 2021. These show that others have already published/presented that bone loss occurs in mouse models of COVID. Thus, their data corroborate those studies and extend these observations. Further, since this is shown in another animal, it adds to the idea that this observation may in fact translate into humans. This should be emphasized and expanded upon. (PMID: 34607050 AND abstract #1025: Blaine Christian presenting author and abstract #LB-1105: Christopher Dalloul presenting author).

Response 1: Thank you for the comment. We agree with the Reviewer and have now cited the published paper (PMID: 34607050) and abstracts as suggested. We have added this in the Discussion as suggested (Line 325, Page 11).

2. The established guidelines in the field for uCT assessments and reporting should be followed (PMID: 20533309). Several of the abbreviations are wrong (Tb.Th – is trabecular thickness) and the methods is not complete and should be expanded for others to reproduce results. SMI and Tb.Sp should be reported as these would be valuable parameters to understand.

Response 2: Thank you for the comment. We have added this reference as well as changed the abbreviations as suggested. We have also provided the detailed methods for the evaluation of bone microstructure in this study as suggested (Line 478, Page 16). To better reflect the changes in trabecular structure after the SARS-CoV-2 infection, we further supplemented data on structure model index (SMI) and trabecular separation (Tb. Sp) in the supplementary information of the revised manuscript (Fig. S1a, c, d).

3. Additional expansion in the discussion on how to reconcile differences in your findings from the literature is required. For example, abstract #1025 shows virus in bone tissue and bone marrow macrophages have also been shown to have virus within them. These references should be added and discussed in the context of the authors findings. <https://www.biorxiv.org/content/10.1101/2021.04.14.439793v2.abstract>

Response 3: Thank you for the comment and the preprint study suggested.

1) We notice that in both abstract #1025 and the preprint (Gao J. et al. 2021), the presence of SARS-CoV-2 in musculoskeletal tissue was confirmed in a lethal transgenic mouse model expressing humanized ACE2 receptors. It was further shown that the SARS-CoV-2 entry to bone marrow macrophages (BMMs) in this model is dependent on the expression of neuropilin-1 (NRP1) rather than the widely recognized receptor ACE2 (Gao J. et al. 2021). Therefore, we are not sure whether the findings in these studies could address the common COVID-19-associated musculoskeletal manifestations in human patients. Indeed, the direct infection of SARS-CoV-2 through ACE2 in bone tissue is very unlikely, because there is little to no expression of *ACE2* or co-expression of *TMPRSS2* (both vital for viral entry) in bone marrow based on single-cell RNA-sequencing data from multiple tissues of human donors (Sungnak W. et al. 2020). A recent study further demonstrated that, SARS-CoV-2 was not able to bind and infect hematopoietic stem and progenitor cells (HSPCs) in bone marrow, from which BMMs were derived (Encabo H.H. et al. 2021).

2) In our study, by using the well-established golden Syrian hamster model, we aim at studying the pathological changes in skeletal system at both the acute

infection phase and the post-recovery phase, which are more relevant to the majority of SARS-CoV-2 infected patients in the real world. In our previous studies, we have shown the viral load in golden Syrian hamster peaked at 4 dpi and gradually reduced to an undetectable level after 7 dpi (Chan J. et al. 2020).

3) Consistent with our findings, it has also been reported by another group that, while the viral load in the lung tissue of SARS-CoV-2-infected hamsters dropped from the peak level at 3 dpi to a significantly lower but still detectable level at 9 dpi, the viral load in bone marrow remained at the same level as mock-infected control over the entire study period (Yang S. et al. 2021). This highlights the consistently absent detection of SARS-CoV-2 in the bone marrow of golden Syrian hamsters.

We have revised the Discussion to include these information as suggested (Line 352, Page12).

Ref:

1. Junjie, Gao, et al. "Neuropilin-1 Mediates SARS-CoV-2 Infection in Bone Marrow-derived Macrophages." bioRxiv (2021).
2. Sungnak, Waradon, et al. "SARS-CoV-2 entry factors are highly expressed in nasal epithelial cells together with innate immune genes." Nature medicine 26.5 (2020): 681-687.
3. Encabo, Hector Huerga, et al. "Human erythroid progenitors are directly infected by SARS-CoV-2: Implications for emerging erythropoiesis in severe COVID-19 patients." Stem cell reports 16.3 (2021): 428-436.
4. Chan, Jasper Fuk-Woo, et al. "Simulation of the clinical and pathological manifestations of coronavirus disease 2019 (COVID-19) in a golden Syrian hamster model: implications for disease pathogenesis and transmissibility." Clinical infectious diseases 71.9 (2020): 2428-2446.
5. Yang, Shiu-Ju, et al. "Characterization of virus replication, pathogenesis, and cytokine responses in syrian hamsters inoculated with SARS-CoV-2." Journal of Inflammation Research 14 (2021): 3781.

4. Additional discussion and clarification of differences in circulating, vs bone tissue, vs cell findings of the interleukins (and other cytokines) discussed should be included. As it currently stands, some elements are confusing to understand after multiple readings of these sections.

Response 4: Thank you for the valuable comments. We have revised our Discussion to avoid the confusion as suggested by the Reviewer (Line 359, Page 12):

“Multiplex quantification of immune mediators of SARS-CoV-2 peptide-stimulated cells from different organs also indicated that SARS-CoV-2–specific immune responses were primarily located in the lung or lung associated lymph nodes instead of the bone marrow (Poon M. et al, 2021). Although we measured a

significant increase in serum level of IL-1 β , the gene expression of IL-1 β in bone tissue did not significantly change. This might indicate that the increase in IL-1 β level in the bone tissue may be primarily attributed to the SARS-CoV-2-induced immune response in the respiratory system. After the inflammatory cytokines (e.g., IL-1 β and TNF- α) produced in respiratory tract reached skeletal tissue via circulation, they quickly modulated the monocyte-macrophage lineage residing there to initiate additional pro-inflammatory cytokine cascade in the bone tissue. The cytokines produced in the skeletal system, including but not limited to IFN- γ , IL-6, and PGE2, not only contributed to osteoclastogenesis in several interdependent signaling pathways (Redlich K. et al, 2012), but also augmented the pro-osteoclastogenic actions of IL-1 β and TNF- α by upregulating the expression of IL-1R1, which was implicated to primarily mediate pathological bone resorption (Trebec D. et al, 2007)”

Ref:

1. Poon, Maya ML, et al. "SARS-CoV-2 infection generates tissue-localized immunological memory in humans." *Science Immunology* 6.65 (2021): eabl9105.
2. Redlich, Kurt, and Josef S. Smolen. "Inflammatory bone loss: pathogenesis and therapeutic intervention." *Nature reviews Drug discovery* 11.3 (2012): 234-250.
3. Trebec, Diana P., et al. "Increased expression of activating factors in large osteoclasts could explain their excessive activity in osteolytic diseases." *Journal of cellular biochemistry* 101.1 (2007): 205-220.

5. It is unclear where all of the samples went or why some figures have different time points etc. For example why is a 60 day mock not included in Figure 1A? Were those not done? Which mock dosing is shown in the figures – it is not clear if 4dpi, 30dpi or what? It would be better to show all of the mock dosing or compare for each time point. That is the proper control for 60 dpi infected is to have a 60 dpi mock-infected. Related to this why was a different volume of vehicle used for the infection? It is known that mock-infection can also cause pneumonia and other lung issues and that needs to be properly accounted for and volume of liquid the mice are inhaling can impact this. These limitations minimally need to be addressed and acknowledged. Likewise, some figures show fewer samples than what appears to be listed in the methods, these differences should be explained.

Response 5: Thank you for the comment. We have now included the individual data points of the mock-infected controls at 4dpi, 30dpi, and 60dpi as suggested by the Reviewer (revised Figures 1a and 1d). We have also included the H&E staining image of 4 dpi, which was missing from the original Fig.1e. We have also corrected the description of the volume used for the SARS-CoV-2-infected groups – each hamster was intranasally challenged with 10⁵ PFU of SARS-CoV-

2 in 50µl (not 100µl) of PBS which was the same volume as the mock-infected groups. We apologize for this unintentional error of method description.

6. In figure 2, why is 4 dpi not shown in panel C, why is 60 dpi not shown in panels d & e, addition of these in the figure and similarly showing these data in the subsequent figures are important to understand the whole picture. For example, in Figure 3 & 4, having the data would demonstrate whether the levels are only up early (during the acute phase) and then decline during the recover phase. This key understanding would significantly improve our understanding of what is occurring with SARS-CoV-2 infection and mechanistically is more important than repeating wild-type osteoclast studies with treatments of cytokines/antibodies. The authors should have these samples based on their study design and other data presented.

Response 6: Thank you for the comments. We have included the H&E staining image of 4 dpi sample in the revised Fig.2c.

Unfortunately, we did not collect the lumbar vertebrae at 60 dpi in our experiments as we initially only focused on the long bones. However, we believe that this would not compromise our findings on the inflammatory bone loss induced by SARS-CoV-2 infection, as the bone loss is evident at both 4 dpi and 30 dpi already. Moreover, we have shown that there is no further significant bone loss between 30 dpi and 60 dpi (Fig. 1d and Fig. 2b). Therefore, to ascertain the judicious use of animals, we did not supplement the 60 dpi data for lumbar vertebra in this work.

Based on our µCT measurement and histology data in this study, we showed that the bone loss is evident primarily at around 4 dpi. This is consistent with our previous studies (Chan J. et al. 2020, Yuan S. et al. 2021, Zhuo D. et al. 2021) and reports by other groups (Imai M. et al. 2020, Yang S. et al. 2021) showing that the severity of SARS-CoV-2 infection in golden Syrian hamsters peaks at about 4 dpi and gradually resolves at or after 7 dpi. To ascertain the judicious use of animals, we focused our analyses in Fig. 3&4 regarding the inflammatory osteoclastogenesis at this prominent stage (4 dpi). We have also added what suggested by the reviewer as a future direction in the Discussion (Line 433, Page 14).

Ref:

1. Chan, Jasper Fuk-Woo, et al. "Simulation of the clinical and pathological manifestations of coronavirus disease 2019 (COVID-19) in a golden Syrian hamster model: implications for disease pathogenesis and transmissibility." *Clinical infectious diseases* 71.9 (2020): 2428-2446.
2. Yuan, Shuofeng, et al. "Clofazimine broadly inhibits coronaviruses including SARS-CoV-2." *Nature* 593.7859 (2021): 418-423.

3. Zhou, Dongyan, et al. "Robust SARS-CoV-2 infection in nasal turbinates after treatment with systemic neutralizing antibodies." *Cell host & microbe* 29.4 (2021): 551-563.
4. Imai, Masaki, et al. "Syrian hamsters as a small animal model for SARS-CoV-2 infection and countermeasure development." *Proceedings of the National Academy of Sciences* 117.28 (2020): 16587-16595.
5. Yang, Shiu-Ju, et al. "Characterization of virus replication, pathogenesis, and cytokine responses in syrian hamsters inoculated with SARS-CoV-2." *Journal of Inflammation Research* 14 (2021): 3781.

7. Many of the mouse cell osteoclast studies which the authors are considering the mechanism of how the cytokine storm is impacting bone loss in their model are studies that others have done over the years and should be appropriately acknowledged.

Response 7: Thank you for the comment. We agree with the Reviewer and have included the appropriate references as suggested in the Discussion (Line 374, Page 13):

"As two of the most significantly upregulated inflammatory cytokines in the serum samples of SARS-CoV-2-infected hamsters, IL-1 β and TNF- α have been known as potent stimulators of bone resorption from as early as the 1980s (Gowen M. et al. 1983; Pfeilschifter J. et al. 1989; Stashenko P. et al. 1987). They are both shown to be essential in RANKL-induced osteoclast formation through the activation of osteoclastogenesis-related signalling pathways such as the c-Jun N-terminal Kinase (JNK) signaling (Lam J. et al. 2000; Lee S. et al. 2006). Besides their interdependent roles in mediating inflammatory osteopenia (Polzer K. et al. 2010), IL-1 β and TNF- α were also found to synergistically interact with other pro-inflammatory cytokine to stimulate osteoclastic differentiation (Stashenko P. et al. 1987; Taki N. et al. 2007)."

Ref:

1. Gowen, Maxine, et al. "An interleukin 1 like factor stimulates bone resorption in vitro." *Nature* 306.5941 (1983): 378-380.
2. Pfeilschifter, Johannes, et al. "Interleukin - 1 and tumor necrosis factor stimulate the formation of human osteoclastlike cells in vitro." *Journal of Bone and Mineral Research* 4.1 (1989): 113-118.
3. Stashenko, P., et al. "Synergistic interactions between interleukin 1, tumor necrosis factor, and lymphotoxin in bone resorption." *The Journal of Immunology* 138.5 (1987): 1464-1468.
4. Lam, Jonathan, et al. "TNF- α induces osteoclastogenesis by direct stimulation of macrophages exposed to permissive levels of RANK ligand." *The Journal of clinical investigation* 106.12 (2000): 1481-1488.

5. Lee, Sun-Kyeong, et al. "RANKL-stimulated osteoclast-like cell formation in vitro is partially dependent on endogenous interleukin-1 production." *Bone* 38.5 (2006): 678-685.
6. Polzer, Karin, et al. "Interleukin-1 is essential for systemic inflammatory bone loss." *Annals of the rheumatic diseases* 69.01 (2010): 284-290.
7. Taki, Naoya, et al. "Comparison of the roles of IL-1, IL-6, and TNF α in cell culture and murine models of aseptic loosening." *Bone* 40.5 (2007): 1276-1283.

8. Your graphical abstract and Figure 8 panel e need to be modified. Why would IL-6 be included in the figure? Your data show that IL-6 serum levels are no different between mock and 4 dpi animals. Your bone RT-PCR data show that IL-6 is down in the bones of 4dpi animals compared to mock-infected animals. So how can your figure say that is part of the pathway? Your mouse cell data suggests it is up but your in vivo infection data does not show that. This contradiction needs to be flushed out in the discussion. And text related to the role of IL-6 needs to be modified accordingly.

Response: Thank you for the comment. We have revised our Graphical Abstract and Figure 8f and deleted IL-6 in these figures as suggested. We also further elaborate our findings in the Discussion (Line 417, Page 14):

“As a major indicator but not the sole driver in the pathology of COVID-19, IL-6 is commonly found significantly increased in patients with exacerbating disease progression (Wang J. et al, 2020). However, many studies have shown that the serum level of IL-6 would not be significantly upregulated in hamsters after SARS-CoV-2 infection (Port J. et al, 2021; Kinoshita T. et al, 2021), because SARS-CoV-2 would not cause severe or even lethal disease in hamsters. Instead, the virus would induce a mild to moderate pathologic procedure resembling the clinical situation found in most COVID-19 patients. Therefore, our observation suggests that the inflammatory bone loss can happen in mild to moderate infection or post-recovery cases of COVID-19 patients.”

Ref:

1. Wang, Jin, et al. "Cytokine storm and leukocyte changes in mild versus severe SARS - CoV - 2 infection: Review of 3939 COVID - 19 patients in China and emerging pathogenesis and therapy concepts." *Journal of leukocyte biology* 108.1 (2020): 17-41.
2. Port, Julia R., et al. "SARS-CoV-2 disease severity and transmission efficiency is increased for airborne compared to fomite exposure in Syrian hamsters." *Nature communications* 12.1 (2021): 1-15.
3. Kinoshita, Takaaki, et al. "Co-infection of SARS-CoV-2 and influenza virus causes more severe and prolonged pneumonia in hamsters." *Scientific Reports* 11.1 (2021): 1-11.

9. Although I agree 3 month mice are younger than 6 month mice, both are considered adults but 6 months is NOT considered middle aged. What was the rationale for using 2 ages of the cells in Figure 7&8, and now how does one explain the differences. More discussion about the differences over time is required for these data as it is difficult to reconcile.

Response: Thank you for the professional comments from the reviewer. In addition to our *in vivo* findings about the inflammatory bone loss in the adult hamsters, we further included *in vitro* experiments using BMMs isolated from 3-month-old or 6-month-old mice, because some symptoms caused by SARS-CoV-2 were found to be age-related. However, we did not include old (aged) mice because they may already suffer from aging-related osteoporosis so the impact of SARS-CoV-2-induced inflammatory bone loss may not be obvious.

3-month-old mouse is widely recognized as young mouse model in many studies, including the ones concerning COVID-19 (Villeda S. et al. 2014; Dinnon K. et al. 2020). Meanwhile, mice at 6-month-old are considered matured adult instead of being noted as “middle-aged” between 10-to-14-month-old (Flurkey K. et al. 2007). Therefore, to better match our *in vivo* study done in adult hamster, we used BMMs isolated from 3-month-old (young adult) and 6-month-old (matured adult). To avoid confusion of the readers, we have revised the text in Results and Discussion.

We have also expanded our discussion as suggested (Line 390, Page 13):

“As we have demonstrated the SARS-CoV-2-induced bone loss in adult hamster, we further involved *in vitro* experiments using BMMs isolated from 3-month-old (young adult) or 6-month-old (matured adult) mice to test whether the inflammatory-mediated osteoclastogenesis is age-related. Importantly, we demonstrated that BMMs isolated from young adult mice were more responsive to the pro-osteoclastic stimulation of IL-1 β . This is clinically relevant because young patients generally have stronger ability to adequately respond to viral infections with rapid production of high level of pro-inflammatory cytokines (Ng P. et al. 2004; Costagliola G. et al. 2021). This heightened pro-inflammatory response, together with the lower baseline pro-inflammatory state in young patients, makes them more susceptible to various syndromes related to immune dysregulation (Costagliola G. et al. 2021; Hobbs C. et al. 2020). Meanwhile, we also found that the involvement of other pro-inflammatory cytokines, such as TNF- α and IFN- γ , contributed to a more prominent effect on promoting the osteoclastic activities in BMMs isolated from matured adult mice than the ones from young mice. This might explain why musculoskeletal symptoms are mostly seen in adult patients rather than children and the elderly (ISARIC Clinical Characterisation. 2021).”

Ref:

1. Villeda, Saul A., et al. "Young blood reverses age-related impairments in cognitive function and synaptic plasticity in mice." *Nature medicine* 20.6 (2014): 659-663.
2. Dinnon, Kenneth H., et al. "A mouse-adapted model of SARS-CoV-2 to test COVID-19 countermeasures." *Nature* 586.7830 (2020): 560-566.
3. Flurkey, Kevin, Joanne M. Curren, and D. E. Harrison. "Mouse models in aging research." *The mouse in biomedical research*. Academic Press, 2007. 637-672.
4. Ng, Pak C., et al. "Inflammatory cytokine profile in children with severe acute respiratory syndrome." *Pediatrics* 113.1 (2004): e7-e14.
5. Costagliola, Giorgio, Erika Spada, and Rita Consolini. "Age - related differences in the immune response could contribute to determine the spectrum of severity of COVID-19." *Immunity, inflammation and disease* (2021).
6. Hobbs, Charlotte V., et al. "COVID-19 in children: a review and parallels to other hyperinflammatory syndromes." *Frontiers in Pediatrics* 8 (2020).
7. Group, ISARIC Clinical Characterisation. "COVID-19 symptoms at hospital admission vary with age and sex: results from the ISARIC prospective multinational observational study." *Infection* (2021): 1.

10. Discussion should include future directions which would according to the focus of the examination of IL-1B in wild-type cells be to perform in vivo infection studies and treat animals with neutralizing antibody and see if bone parameters improve and osteoclast numbers are reduced. The data presented are all hypothetical and correlative, not truly mechanistic as in the infection model no interventions were provided to test whether the cytokine storm is truly responsible. There are places throughout the manuscript where the interpretation is stretched in this regard. Completion of this study and showing an improvement in bone indices is truly the best mechanistic study for this manuscript. Last paragraph of the introduction needs revising as not accurate, same in the abstract and discussion. Carefully only stating what you have shown is critical.

Response 10: Thank you for the comment.

1) We have included the future direction of this study in the Discussion as suggested (Line 433, Page 14):

"In a recently published meta-analysis, Anakinra, a recombinant IL-1 receptor antagonist, was suggested as a safe, anti-inflammatory treatment option to reduce the mortality risk in patients admitted to hospital with moderate to severe COVID-19 pneumonia, especially in the presence of signs of hyperinflammation. This suggests that adequate use of anti-inflammatory treatment may be beneficial for COVID-19 during both the acute phase and in the chronic phase involving long-term inflammatory complications. Therefore, evaluation for the

effects of anti-inflammatory agents such as IL-1 receptor antagonists and IL-1 neutralizing antibodies on the prevention and/or treatment of SARS-CoV-2 infection-induced inflammatory bone loss should be included in future studies.”

2) We have also revised the last paragraph of the introduction, abstract and discussion to make sure there is no over-claim as suggested by the reviewer.

11. I am not sure that the idea of a second wave is accurate. If others have documented this please refer to those data. Second suggests difference over time and I am not clear with the data shown that it is not all from the same insult – systemic cytokine storm, all in the first wave to use your terminology. Clearly, modifying Figure 8e based on whether this is your hypothesis or accurate may be needed.

Response 11: Thank you for the comment. We agree with the reviewer that the term “second wave” maybe not be the best description and have therefore changed the it to “pro-inflammatory cascade” throughout the manuscript, which is more widely used in the literature.

Minor Concerns.

1. There are some grammatical errors which should be carefully checked during revisions for ease of reading. (line 173 as example)

Response 1: Thank you for the comment. We have revised the sentence into “In addition to the increase in the number of TRAP+ osteoclasts, we also found there were significantly more osteoclast progenitors, including CD68+ macrophages and RANK+ preosteoclasts, after SARS-CoV-2 infection”

Additionally, we have also checked the manuscript thoroughly again to correct other grammatical errors.

2. Western, in Western blot should be capitalized throughout (including supplementary materials).

Response 2: Thank you for the comment. We have revised these accordingly as suggested.

3. Unless published does not allow, all figures should discuss age and sex of hamsters so reader isn’t left waiting for methods to understand – or at least write early in results. In fact Figure 1A could have this information and how many hamsters are in each group to make the study design more clear.

Response 3: Thank you for the comment. We have included the information concerning the number and/or gender of the hamsters in the figure legends where possible as suggested.

4. Figure 5 panel A and b should be revised for clarity – panel A should have label on rows with Mock and 4dpi. Fig 5b should be all red squares since all 4 dpi (I presume, but this needs clarification).

Response 4: Thank you for the comment. We have revised these accordingly as suggested.

Reviewer #2 (Remarks to the Author):

In this study entitled “SARS-CoV-2 infection induces inflammatory bone loss in golden Syrian hamsters”, the authors reported the novel findings of pathological bone loss after SARS-CoV-2 infection in the Syrian hamster model. They showed that SARS-CoV-2 infection consistently causes significant bone loss of bone trabeculae in both the long bones (femur and tibia) and lumbar vertebrae of the hamsters. They serially studied the bone losses and showed that the changes progressively worsened from the acute phase at 4 days post-infection to the post-recovery phase at 30-60 dpi. The authors then investigated the underlying mechanism of these SARS-CoV-2-induced bone losses and showed that virus-induced cytokine dysregulation upregulated osteoclastic differentiation of monocyte-macrophage lineage; whereas the osteoblastic differentiation was not significantly altered.

The findings of this work is novel, important, and timely especially as “long COVID-19” or “chronic COVID-19” complications are increasingly being reported in patients who have recovered from the initial infection. The team in this work has excellent track records in coronavirus / COVID-19 research and is the group that reported the first established Syrian hamster model for COVID-19 (Chan JF et al., Clin Infect Dis 2020). The methods and results are well described and the figures are of high quality to illustrate the findings. The authors’ proposed reasons why these bone changes may have been overlooked in patients (in the Introduction) are reasonable and clinically relevant. Coincidentally, the Reviewer noted that a very recent paper describing bone losses in mice was published on 2nd October 2021 (Awosanya OD et al., Bone 2021) after the current work was submitted to Nature Communications. However, the Reviewer finds that the current submission is superior to the published paper for a number of major reasons:

1) The animal model used in the current submission (Syrian hamster) is well-established and closely mimics mild-to-moderate infection in human. Comparatively, the animal model using K18-hACE2 transgenic mice in the Bone paper is a partially to fully lethal model (depending on virus inoculum) and does not suitably represent the majority of patients’ mild-to-moderated COVID-19 clinically. This is especially clinically relevant because long COVID-19 complications occur in surviving patients and not those who have already succumbed during the acute phase.

2) Phenotypically, the current study is more comprehensive which describes the multifocal involvement as well as serial changes in the skeletal system of the infected vs mock-infected hamsters. These data on different bones and longitudinal effects at different time points after SARS-CoV-2 infection were not available in the Bone paper.

3) There was little, if any, mechanistic data provided in the Bone paper (only provided a “hypothetical model” without any experimental data to support the proposed mechanism). Comparatively, the authors in this study used a number

of well-designed in vitro and in vivo experiments to demonstrate SARS-CoV-2-induced cytokine storm and the subsequent immune dysregulation in bone microenvironment as the underlying mechanism contributing to the osteoclastogenesis. In particular, this study identified IL-1 β and its receptor IL-1R1 as the key signaling involved in inflammatory SARS-CoV-2-induced bone loss, which may further serve as a potential therapeutic target for COVID-19. Overall, the findings in this study are important and timely. A number of comments / clarifications listed below should be addressed before further consideration for publication:

Response: We thank the reviewer for the encouraging comments.

Major comments:

1. The decrease in bone density happens during the aging process. Can the difference in bone density caused by the natural aging process? It would be important to clearly show the accurate age of the hamsters involved in this study. Is the age of hamsters in mock control group same as the one in infected group? Please specifically state the ages of the hamsters in the Methods or Results.

Response 1: Thank you for the comment. To demonstrate that our SARS-CoV-2-infected and mock-infected hamsters were age-matched, we have now revised the way we present our data in Figures 1 and 2.

2. The study demonstrates significant loss of bone trabeculae, but what about the cortical bone? Is there any change (thickness or density) after the infection of SARS-CoV-2?

Response 2: Thank you for the comment. We did not show the data concerning the comparison of cortical bone between SARS-CoV-2 infected hamsters and PBS-treated (mock) hamsters in the original version of the manuscript because they were not significantly different. However, we agree with the reviewer that this information may be valuable for the readers to know that the bone resorption primarily happens in cancellous bone tissues. Therefore, we have now added provided these data in Supplementary information Fig.S1b as suggested.

3. To ensure the robustness of viral antigen detection, have the authors tried more than one antibody for the detection of viral nucleocapsid protein?

Response 3: Thank you for the comment. We used two different antibodies targeting SARS-CoV-2 nucleocapsid protein. The first antibody is an in-house developed antibody by our team (Liu L. et al. 2020). The second antibody is a commercial product (MA1-7403) from ThermoFisher, which has also been verified in several publications (Subramanian B. et al. 2021; Wu C. et al. 2021).

Additionally, we have also used RT-qPCR to further verify our findings (Fig. 5b) which was in agreement with our antigen detection tests.

Ref:

1. Liu, Lihong, et al. "Potent neutralizing antibodies against multiple epitopes on SARS-CoV-2 spike." *Nature* 584.7821 (2020): 450-456.
2. Subramaniyan, Bharathiraja, et al. "Characterization of the SARS-CoV-2 Host Response in Primary Human Airway Epithelial Cells from Aged Individuals." *Viruses* 13.8 (2021): 1603.
3. Wu, Chien-Ting, et al. "SARS-CoV-2 infects human pancreatic β cells and elicits β cell impairment." *Cell metabolism* (2021).

4. It is suggested that the infected hamsters generally recovered at about 7 to 10 dpi, however, it seems the bone loss primarily happen between 4 to 30 dpi, why?

Response 4: Thank you for the comment. Although the acute features in the respiratory tract resolve from 7 dpi with decreasing virus titer, the cytokine storm-induced pathological changes in multiple organs can last for much longer. The persistence of symptoms in those who have recovered from SARS-CoV-2 infection has been extensively reported and termed as "Long COVID". In this study, we showed the osteoclastogenesis is not directly induced by the virus infection in bone tissue. Instead, it is induced by the proinflammatory cytokines originally produced in the respiratory system. Moreover, the cytokine storm also leads to an amplified and broadened pro-inflammatory cascade in the musculoskeletal system, resulting in the persistence of bone resorption after the acute inflammation stage. Meanwhile, it is notable that the lifespan of osteoclasts is around 2 weeks so once the osteoclastic differentiation is initiated at 4 dpi, the bone resorption may last for a longer period of time.

5. Patients with long COVID-19 may have muscle and/or joint pain. Can the authors show data on whether virus infection in this Syrian hamster model?

Response 5: Thank you for the comment. The wide observation on muscle and joint pain in long COVID-19 has led to the question whether there is virus infection in these areas. We first used immunofluorescent staining to show that there is no SARS-CoV-2 nucleocapsid protein in the muscle or joint, despite the high-level expression of the ACE2 (Fig.5a). Moreover, we verified our antigen detection test findings using RT-qPCR which also showed that viral RNA was undetectable in the long bone tissues, including the femur, tibia, knee joint, and attached muscle. These findings are consistent with reports elsewhere showing there is no evidence of virus in muscle (Pitscheider L. et al. 2021) or joint (Grassi M. et al. 2021).

Ref:

1. Pitscheider, Lea, et al. "Muscle involvement in SARS-CoV-2 infection." European journal of neurology 28.10 (2021): 3411-3417.
2. Grassi M, Giorgi V, Nebuloni M, et al AB0671 No evidence of sars-cov-2 in the knee joint: a cadaver study. Annals of the Rheumatic Diseases 2021

Minor comments:

1. What do the arrow heads indicate in Figures 4b, 4d, and 5a?

Response 1: Thank you for the comment. The arrow heads in Fig.4b indicate CD68+ and TRAP+ osteoclasts. The arrow heads in Fig. 4d and Fig. 5a indicate osteoclasts on bone surface. We have revised our figure legends accordingly as suggested.

2. If only IL-1 β neutralizing antibody was used, it should be Neu-Ab instead of Neu-Abs.

Response 2: Thank you for the comment. We have revised these accordingly throughout the manuscript.

3. Quantification of WB in Fig. 8a should be provided.

Response 3: Thank you for the comment. We have added the quantitation of WB (Fig. 8a) in the new Fig.8b.

4. In Fig. S3c, it should be NF κ B (should be consistently spelt as NF- κ B).

Response 4: Thank you for the comment. We have revised this as suggested.

5. Typos: antii-RANK (Abcam, ab13918) in the Method part.

Response: Thank you for the comment. We have revised this as suggested.

Reviewer #3 (Remarks to the Author):

Yeung et al. describe inflammatory bone loss in golden Syrian hamster model induced by SARS-CoV-2 infection. They show convincing micro-CT scans in different bones and perform experiments of gene expression at the RNA and protein levels (RT-qPCR, western blot, immunofluorescence, etc) in hamster or tissues or mouse BMMs. They find that there is a first and a second wave of cytokine storm mediated by IL-1B and other cytokines, the second wave being the main responsible for the bone loss. Although the experiments described are elegant and the importance of the findings is high, the paper needs a better discussion of the mechanism proposed – that IL-1beta plays an important role in bones loss through cytokine storm. While the mouse experiments appear to hint at this mechanism, the levels of recombinant IL1-beta in this experiment are likely much higher than the increase observed in hamsters (~25%).

Response: We thank the reviewer for the generally positive comments.

- Do the authors think that an increase in IL1-B of about 25% (Fig 5c) would be able to explain the phenotypes observed? In the experiments of figures 7 and 8, by how many fold was the expression of IL-1beta increased in the presence of recombinant IL-1beta?

Response 1: Thank you for the valuable comment. As one of the strongest stimulators for bone resorption, IL-1 β can significantly promote the formation of osteoclasts even at a very low concentration (i.e. 2.5×10^{-13} M) (Stashenko P. et al. 1987). The effects of IL-1 β on osteoclastogenesis would become more significant in the presence of other pro-inflammatory cytokines, such as TNF- α , because they have synergetic effects on osteoclastic differentiation (Stashenko P. et al. 1987; Taki N. et al. 2007). We showed that the cytokine originating from the respiratory tract also led to an amplified and broadened pro-inflammatory cascade in the musculoskeletal system, leading to the upregulation of bone resorption. Indeed, the level of IL-1 β in bone was increased by 4~6-fold according to immunofluorescent staining (Fig. 6a) and Western blot (Fig.S5a).

For the experiments of Fig.7 and 8, 1 ng/mL recombinant murine IL-1 β was added to the culture medium. Considering the background level of IL-1 β in RANKL induced BMM culture to be around 1.5 ng/mL (Akbar M. et al. 2017), the recombinant IL-1 β added only contributed to a less than 2-fold increase in this cytokine, which is commonly observed during inflammatory disease.

Ref:

1. Stashenko, P., et al. "Synergistic interactions between interleukin 1, tumor necrosis factor, and lymphotoxin in bone resorption." *The Journal of Immunology* 138.5 (1987): 1464-1468.

2. Taki, Naoya, et al. "Comparison of the roles of IL-1, IL-6, and TNF α in cell culture and murine models of aseptic loosening." *Bone* 40.5 (2007): 1276-1283.
3. Akbar, Mohammad Ahsanul, et al. " α -1 Antitrypsin Inhibits RANKL-induced Osteoclast Formation and Functions." *Molecular Medicine* 23.1 (2017): 57-69.

- The authors postulate that a cytokine storm leads to bone loss. However, on figure 5c, the increase noted in IL-1-beta and TNF-alpha seem very small to be considered as "cytokine storm". Cytokine storm usually means a large increase in cytokines in a short period of time. Further, only a few cytokines/chemokines are measured, and most are shown as difference in gene expression rather than protein levels. How do the authors support the conclusion of a cytokine storm with so few measurements?

Response 2: Thank you for the comment. We and others have previously demonstrated that SARS-CoV-2-infected hamsters develop cytokine/chemokine dysregulation which is most prominently observed in the lungs (Chan JF. et al 2020; Yang S. et al. 2021). The striking similarities in the pro-inflammatory response between hamster model and human patients have been confirmed by single-cell RNA sequencing (Nouailles G. et al. 2021). In this work, we also detected significantly increased IL-1-beta and TNF-alpha at the protein level (Fig. 5c) in serum, although to a lower intensity than the lung as expected. It is important to note that the Syrian hamster COVID-19 model represents mild to moderate human disease and therefore is physiologically relevant to represent most non-lethal COVID-19 human cases, and that the cytokine/chemokine levels would not be expected to be a "lethal" type of cytokine storm. However, it is generally accepted in the field that this is still considered a "cytokine storm" (Yang S. et al. 2021; Lu M. et al. 2021; Sahoo D. et al. 2021). Nevertheless, we agree with the reviewer and have now used the term "cytokine dysregulation" instead of "cytokine storm". As for the cytokine/chemokines, we selected IL-1 β , TNF- α , and IL-6, which are known as three of the most important pro-inflammatory cytokines of the innate immune response, because there are very limited test assays available for hamsters (unlike mice). We have included this as a limitation in our manuscript (Line 440, Page 15).

Ref:

1. Chan, Jasper Fuk-Woo, et al. "Simulation of the clinical and pathological manifestations of coronavirus disease 2019 (COVID-19) in a golden Syrian hamster model: implications for disease pathogenesis and transmissibility." *Clinical infectious diseases* 71.9 (2020): 2428-2446.
2. Yuan, Shuofeng, et al. "Clotrimazole broadly inhibits coronaviruses including SARS-CoV-2." *Nature* 593.7859 (2021): 418-423.
3. Nouailles, Geraldine, et al. "Temporal omics analysis in Syrian hamsters unravel cellular effector responses to moderate COVID-19." *Nature communications* 12.1 (2021): 1-18.

4. Yang, Shiu-Ju, et al. "Characterization of virus replication, pathogenesis, and cytokine responses in syrian hamsters inoculated with SARS-CoV-2." *Journal of Inflammation Research* 14 (2021): 3781.
5. Lu, Mijia, et al. "A safe and highly efficacious measles virus-based vaccine expressing SARS-CoV-2 stabilized prefusion spike." *Proceedings of the National Academy of Sciences* 118.12 (2021).
6. Sahoo, Debashis, et al. "AI-guided discovery of the invariant host response to viral pandemics." *EBioMedicine* (2021): 103390.

- What is the rationale for using only male hamsters during these studies when there are known sex differences on bone density and health.

Response 3: Thank you for the comment. We agree with the reviewer and have included data on female hamsters. The data shown in Fig.1 and 2 suggested the bone loss happens equally in male and female hamsters.

Minor comments:

- The authors do immunofluorescence in hamster tissue (such as figure 3E), which can be technically difficult given the relative non-cross reactivity between hamsters and other rodents. Can the authors explain better in the text if they tested many antibodies or optimized a special protocol? This would increase rigor and reproducibility between hamster models of SARS-CoV-2 infection models.

Response 4: Thank you for the comment. We recognize the potential issue raised by the reviewer. To address this, we first considered the antibodies already verified in hamster by manufacturers, such as anti-NF- κ B p65 (D14E12) from CST, and the antibodies verified in previous publications, such as anti-SARS-CoV-2 Nucleocapsid Protein (MA1-7403) from ThermoFisher. However, due to the limitation in available antibody for hamsters, we have to use some of the antibodies that have not been tested in hamster model before. For these antibodies, the following principles were followed:

1. We have ensured the target proteins show >90% homology at the amino acid level with the other rodents and was tested for cross-reactivities previously (Zivcec M. et al. 2011);
2. All antibodies used have been verified in other rodent animal models in our previous studies (Qiao W. et al. 2021);
3. The specificity of these antibodies was tested using Western blots before their being used for immunofluorescent staining;
4. Negative control was used in each experiment to exclude possible non-specific staining;
5. The result of the immunofluorescent staining was confirmed by RT-qPCR using verified primers.

The immunofluorescent staining was performed using a standard protocol described in Methods section. For some target protein we tried different antibodies but only reported the verified ones in the Methods section.

Ref:

1. Qiao, Wei, et al. "Sequential activation of heterogeneous macrophage phenotypes is essential for biomaterials-induced bone regeneration." *Biomaterials* 276 (2021): 121038.
2. Zivcec, Marko, et al. "Validation of assays to monitor immune responses in the Syrian golden hamster (*Mesocricetus auratus*)." *Journal of immunological methods* 368.1-2 (2011): 24-35.

- Authors use the expression “challenged with PBS”, but PBS is not a challenge, so this should be corrected.

Response 5: Thank you for comment and we have revised this according to the reviewer’s suggestion.

- I did not see if the authors mention anywhere in the paper: was it previously known that IL1-beta can promote osteoclastogenesis?

Response 6: Thank you for the comment. We have extended our Discussion about previous findings on the pro-osteoclastic effect of IL-1 β in the revised manuscript (Line 374, Page 13):

“As two of the most significantly upregulated inflammatory cytokines in the serum samples of SARS-CoV-2-infected hamsters, IL-1 β and TNF- α have been known as potent stimulators of bone resorption from as early as the 1980s (Gowen M. et al. 1983; Pfeilschifter J. et al. 1989; Stashenko P. et al. 1987). They are both shown to be essential in RANKL-induced osteoclast formation through the activation of osteoclastogenesis-related signalling pathways such as the c-Jun N-terminal Kinase (JNK) signaling (Lam J. et al. 2000; Lee S. et al. 2006). Besides their interdependent roles in mediating inflammatory osteopenia (Polzer K. et al. 2010), IL-1 β and TNF- α were also found to synergistically interact with other pro-inflammatory cytokine to stimulate osteoclastic differentiation (Stashenko P. et al. 1987; Taki N. et al. 2007).”

Ref:

1. Gowen, Maxine, et al. "An interleukin 1 like factor stimulates bone resorption in vitro." *Nature* 306.5941 (1983): 378-380.
2. Pfeilschifter, Johannes, et al. "Interleukin - 1 and tumor necrosis factor stimulate the formation of human osteoclastlike cells in vitro." *Journal of Bone and Mineral Research* 4.1 (1989): 113-118.

3. Stashenko, P., et al. "Synergistic interactions between interleukin 1, tumor necrosis factor, and lymphotoxin in bone resorption." *The Journal of Immunology* 138.5 (1987): 1464-1468.
4. Lam, Jonathan, et al. "TNF- α induces osteoclastogenesis by direct stimulation of macrophages exposed to permissive levels of RANK ligand." *The Journal of clinical investigation* 106.12 (2000): 1481-1488.
5. Lee, Sun-Kyeong, et al. "RANKL-stimulated osteoclast-like cell formation in vitro is partially dependent on endogenous interleukin-1 production." *Bone* 38.5 (2006): 678-685.
6. Polzer, Karin, et al. "Interleukin-1 is essential for systemic inflammatory bone loss." *Annals of the rheumatic diseases* 69.01 (2010): 284-290.
7. Taki, Naoya, et al. "Comparison of the roles of IL-1, IL-6, and TNF α in cell culture and murine models of aseptic loosening." *Bone* 40.5 (2007): 1276-1283.

- The IL-1B neutralizing antibody sometimes has a similar effect as the IL-1B, such as in panels 8A, 7B and 7C (6 months). Is this expected? The authors should discuss these findings.

Response 7: Thank you for the comment. We noticed the neutralizing antibody failed to block the late osteoclastic differentiation especially in BMMs from young mice, as the size of osteoclasts and the expression of CTSK did not drop to the baseline level after the addition of IL-1 β neutralizing antibody. These data suggest that multiple neutralizing antibodies targeting different pro-inflammatory cytokines may need to be considered for limiting IL-1 β -induced osteoclastogenesis. Because we have demonstrated that the initial stimulation of IL-1 β would lead to an amplified and broadened pro-inflammatory cascade, resulting in the production of a variety of cytokines favoring osteoclastogenesis. Therefore, as suggested elsewhere (Kimble R. et al. 1995), simultaneous block of IL-1 β and other pro-inflammatory cytokines is required for preventing bone loss.

Kimble, Robert B., et al. "Simultaneous block of interleukin-1 and tumor necrosis factor is required to completely prevent bone loss in the early postovariectomy period." *Endocrinology* 136.7 (1995): 3054-3061.

- On line 198, the authors say that viral RNA was not detected in bone tissue. But on figure 5b the viral RNA is quantified in bone tissue. Please clarify.

Response 8: Thank you for the comment. To avoid confusing the readers, we have now revised Fig. 5b with the data below the limit of detection at 10^{-2} virus genome copy per beta-actin (log10) being segmented.

- Figure 5: How can the authors explain the differences in IL-6 on figures 5c (trending to increase but n.s.) and 5e (decreasing)?

Response 9: Thank you for the comment. The serum concentration of IL-6 in Fig. 5c represents the circulating cytokine level. The marginal increase in IL-6 in the serum primarily attributes to the SARS-CoV-2 infection in respiratory tract. However, the RT-qPCR data in Fig. 5e determined the expression of gene encoding IL-6 in bone tissue. Since there is no direct virus infection in the skeletal system, the expression of IL-6 was not upregulated in bone. Instead, monocytes, the key producer of IL-6 in bone marrow (Norelli M. et al. 2018), are recruited to the infected respiratory tissues via the bloodstream to mediate antimicrobial activities (Shi C. et al. 2011; Martinez F. et al. 2020). Therefore, the difference in the trends of IL-6 may be explained by the emigration of monocytes in response to SARS-CoV-2 infection.

Ref:

1. Norelli, Margherita, et al. "Monocyte-derived IL-1 and IL-6 are differentially required for cytokine-release syndrome and neurotoxicity due to CAR T cells." *Nature medicine* 24.6 (2018): 739-748.
2. Shi, Chao, and Eric G. Pamer. "Monocyte recruitment during infection and inflammation." *Nature reviews immunology* 11.11 (2011): 762-774.
3. Martinez, Fernando O., et al. "Monocyte activation in systemic Covid-19 infection: Assay and rationale." *EBioMedicine* 59 (2020): 102964.

- In figure S4, panel c p-JNK does not seem to be changed. In this case, it would be better to say “phosphorylated JNK” instead of “phosphorylation of JNK”, for improved clarity.

Response 9: Thank you for the comment. We have revised this as suggested by the reviewer.

- On line 420, why do the authors use only 50 microliters of PBS when 100 microliters of SARS-2 were used?

Response 10: Thank you for the comment. We have corrected the description of the volume used for the SARS-CoV-2-infected groups – each hamster was intranasally challenged with 10^5 PFU of SARS-CoV-2 in 50 μ l (not 100 μ l) of PBS which was the same volume as the mock-infected groups. We apologize for this unintentional error of method description.

Reviewers' Comments:

Reviewer #1:

Remarks to the Author:

The authors did an excellent job incorporating the reviewer feedback. I am satisfied with the revisions and their responses.

The only minor update I might recommend is to have the authors update the statistics in the first paragraph of the introduction to reflect current statistics rather than Aug 2021. But that is not critical.

Reviewer #2:

Remarks to the Author:

The authors have satisfactorily addressed all concerns and revised the manuscript accordingly. Therefore, I recommend publication of this revised manuscript in Nature Communication.

Reviewer #3:

Remarks to the Author:

The authors have carefully and thoughtfully considered all of the reviewer's critiques. The manuscript is much stronger as a result.

Minor comment: While cytokine storm was changed to cytokine dysregulation, it still shows as cytokine storm in the graphical abstract.

2022-03-05

We really appreciate the additional comments from the reviewer and have revised the manuscript accordingly with all the changes highlighted.

REVIEWER COMMENTS:

Reviewer #1 (Remarks to the Author):

The authors did an excellent job incorporating the reviewer feedback. I am satisfied with the revisions and their responses.

The only minor update I might recommend is to have the authors update the statistics in the first paragraph of the introduction to reflect current statistics rather than Aug 2021. But that is not critical.

Response: We thank the reviewer for the encouraging comments. As suggested, we have updated the statistics in the first paragraph showing there have been nearly 346 million cases and nearly 5.5 million deaths as of 23 January 2022.

Reviewer #2 (Remarks to the Author):

The authors have satisfactorily addressed all concerns and revised the manuscript accordingly. Therefore, I recommend publication of this revised manuscript in Nature Communication.

Response: We deeply appreciate the positive comments from the reviewer.

Reviewer #3 (Remarks to the Author):

The authors have carefully and thoughtfully considered all of the reviewer's critiques. The manuscript is much stronger as a result.

Minor comment: While cytokine storm was changed to cytokine dysregulation, it still shows as cytokine storm in the graphical abstract.

Response: We thank the reviewer for the encouraging comments. We apologize for the mistake in the graphical abstract and have revised it accordingly.